# VIBE: VISION TRANSFORMER BASED EXPERTS NETWORK FOR SSVEP DECODING

## ABSTRACT

Steady-state visual evoked potential based brain–computer interfaces (SSVEP-BCIs) have attracted wide attention for their high information transfer rate (ITR) and non-invasiveness. However, existing deep learning methods for SSVEP-BCI decoding have reached a performance bottleneck, as they struggle to fully extract the complex neural signal features required for robust performance. Motivated by advances in vision and time series modeling, here we present a **VI**sion Transformer **B**ased **E**xpert network (VIBE), a multistage deep learning framework for SSVEP classification. VIBE integrates a Vision Transformer (ViT) module to generate rich spatiotemporal representations with data and network enhancement modules in a decoder for frequency recognition. We evaluate VIBE on two large benchmark datasets, including the Benchmark and the BETA dataset spanning 105 subjects. Notably, with just 0.4 seconds of stimulation, our VIBE achieves an ITR of 263.8 bits per minute (bpm) and 202.7 bpm on the Benchmark and BETA datasets, respectively. Experimental results demonstrate that VIBE consistently outperforms state-of-the-art baselines in offline experiments, highlighting its effectiveness as a high-performance decoding strategy for individually calibrated SSVEP-BCIs.

## 1 INTRODUCTION

Brain–computer interfaces (BCIs) provide a direct communication pathway between the brain and external devices, enabling interaction without relying on neuromuscular activity. BCIs have emerged as effective tools for augmentative communication and human-machine interaction, with broad potential applications ranging from neuroprosthesis (Willett et al., 2021; 2023) to the next-generation form of human-computer interaction (Gao et al., 2025). Among noninvasive paradigms, steady-state visual evoked potential based BCIs (SSVEP-BCIs) stand out for their non-invasiveness, high ITR, robustness, and scalability. SSVEPs are frequency-tagged neural responses that can be evoked by periodic visual stimulation, including flickering squares, reversing checkerboards, and moving gratings, and they are elicited over occipital cortex at the stimulation frequency and its harmonics. These frequency-tagged responses exhibit a high signal-to-noise ratio (SNR), enabling SSVEP-BCIs to implement high-speed spellers (Chen et al., 2015b), robotic control, and smart home systems. However, achieving high decoding accuracy under short time windows remains challenging in learning effective neural representations from noisy, data-constrained EEG recordings with complex spatiotemporal and spectral dynamics.

Advancements in SSVEP-BCI decoding have been driven by both traditional linear methods and emerging deep learning models. Early approaches such as canonical correlation analysis (CCA) (Bin et al., 2009) and its filter-bank extension (FBCCA) Chen et al. (2015a) established training-free plug-and-play frequency recognition, while subsequent training-based methods like task-related component analysis (TRCA) Nakanishi et al. (2017) and task-discriminant component analysis (TDCA) (Liu et al., 2021b) designed sophisticated spatial filters using individually calibrated data to significantly boost the decoding performance. However, these linear methods remain limited in capturing the nonlinear and hierarchical patterns of EEG signals. To mitigate these issues, convolutional neural networks (CNNs) and, more recently, Transformer-based architectures have been explored for learning richer spatio-temporal representations from EEG, achieving improvements over traditional baselines (Li et al., 2020; Song et al., 2022). However, both families of models exhibit concrete gaps when applied to SSVEP decoding. CNNs depend on fixed, small receptive fields, which lim-

its their ability to capture long-range temporal structure and harmonic relationships across wider time-scales features that are essential for differentiating densely spaced SSVEP frequencies. While Transformers are in principle capable of modeling global dependencies, existing works primarily deploy them in leave-one-subject-out (LOSO) evaluations, where broad cross-subject scenario is beneficial. In individually calibrated settings, however, this line of research often struggles to extract the fine-grained, local frequency-specific patterns required for high-precision decoding, and typically underperforms despite its expressive capacity.

To overcome these limitations, we introduce **VIBE** (Vision Transformer Based Experts Network), a multistage framework that unifies transformer-based sequence modeling with expert-driven specialization. VIBE employs a ViT module to perform temporal generation, expanding short input sequences into richer representations that preserve multi-scale temporal dependencies. It integrates a Mixture of Experts (MoE) decoder, where experts specialize in different subband–channel–temporal dynamics, and a load-balancing loss ensures diverse expert utilization for better generalization. On top of these architectural innovations, VIBE employs a staged training scheme that progressively pretrains and fine-tunes the ViT-based temporal generation and MoE-based decoding modules, adapting from population data to subject-specific dynamics. It further integrates data augmentation strategies, including temporal stitching, channel chunk shuffling, random temporal cropping, and decorrelation, to regularize training and enhance representation learning for EEG.

We evaluate VIBE on two large benchmark datasets, the Benchmark and the BETA datasets, spanning 105 subjects. Results show that VIBE consistently outperforms both classical and deep learning baselines, achieving state-of-the-art accuracy and ITR under short time windows. These findings establish VIBE as an effective decoding strategy for high-throughput SSVEP-BCIs.

In summary, our main contributions are threefold:

1. A novel hybrid framework that combines ViT-based temporal generation with MoE-based subband-channel-temporal decoding enhancement for SSVEP classification.

2. A staged training scheme that progressively adapts temporal generation and decoding modules from population to subject-specific data.

3. A suite of data augmentation methods designed for SSVEP data, improving robust representation learning from limited and noisy neural data.

## 2 RELATED WORK

**Traditional methods.** Early research focused on traditional frequency recognition methods, which can be broadly divided into training-free and training-based approaches. Canonical correlation analysis (CCA) (Bin et al., 2009) and its filter-bank extension (FBCCA) (Chen et al., 2015a) became widely used due to their plug-and-play traning-free capability. Prior studies using individually calibrated data introduced multiset CCA (MsetCCA) (Zhang et al., 2014), L1-regularized multiway CCA (L1MCCA) (Zhang et al., 2013), and extended CCA (eCCA) (Nakanishi et al., 2014), which improved robustness by leveraging richer reference structures and regularization strategies. More sophisticated spatial filter based methods were developed to further boost accuracy. Task-related component analysis (TRCA) (Nakanishi et al., 2017) significantly improved SSVEP decoding by maximizing trial-to-trial reproducibility. To further address redundancy in TRCA's ensemble design, task-discriminant component analysis (TDCA) (Liu et al., 2021b) eliminated the training of spatial filters class by class and leveraged spatio-temporal neural dynamics, making it a state-of-the-art method for enabling high-speed brain spellers. Despite these advances, traditional methods remain linear and limited in their ability to capture nonlinear, hierarchical representations of EEG.

**Deep learning methods.** Motivated by these limitations, recent research has turned toward learning complex representations from noisy signals in an end-to-end manner. Convolutional neural networks (CNNs) enabled data-driven feature extraction, analogous to filtering in EEG signal processing, and advances such as convolutional correlation analysis (ConvCA) (Li et al., 2020) and deep neural network classifiers (Guney et al., 2022) have surpassed linear baselines. Extensions incorporated fixed and dynamic template networks (Xiao et al., 2022), bidirectional Siamese correlation networks (Zhang et al., 2022), and multiscale CNNs with squeeze-and-excitation blocks (Jin et al., 2024). More recently, Transformer-based architectures leveraging self-attention to capture long-range temporal dependencies have been applied to EEG (Song et al., 2022; Wan et al., 2023),

including SSVEPformer (Chen et al., 2023), DG-Conformer (Liu et al., 2024), SSVEPPoolformer (Li et al., 2025a) and MTSNet (Lan et al., 2025) for cross-subject SSVEP classification. Hybrid approaches such as TRCA-Net (Deng et al., 2023) and discriminant compacted network (Li et al., 2025b) combine spatial filters with neural networks, while ConsenNet (Zhang et al., 2024) leverage a teacher-student framework to further improve performance. Most recently, Mamba-based models such as SUMamba (Dong et al., 2026) integrated multi-scale feature fusion to facilitate classification. However, CNNs remain limited in capturing global dependencies, and Transformers often neglect inductive biases specific to EEG, leaving the representation learning problem unresolved for high-throughput SSVEP-BCIs.

## 3 METHOD

We first define the notation used throughout this work. The multi-channel EEG signal is represented as $X \in \mathbb{R}^{B \times C \times T}$, where $B$ denotes the number of filter banks, $C$ the number of EEG electrodes (channels), and $T$ the total number of sampled timestamps. In our experiments, we consider $B = 3$ sub-bands extracted by band-pass filtering, and $C = 9$ channels selected from classical montage for SSVEP classification (Chen et al., 2015b).

Fig. 1 is not so auto-explicative. This reviewer believe that there is too much text in it, without a clear indication about where to start looking at. Moreover, the caption only talks about points a) and e), but not b-d). I suggest redesigning the figure and its caption entirely.

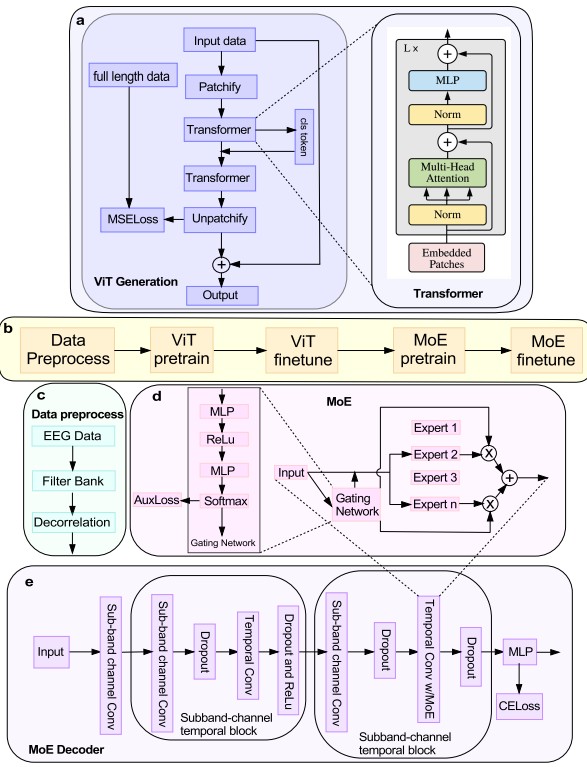

Figure 1: Framework of the proposed VIBE method. Panel (b) presents the overall pipeline, from short SSVEP EEG segments to final classification. Panel (c) depicts the data preparation strategy used during training and testing. Temporal sequence generation with the ViT is detailed in (a), identical for both pretrain and fine-tune stages. The MoE decoder is shown in (e), also identical for both pretrain and fine-tune stages. Panel (d) illustrates the final convolutional temporal layer integrated with the MoE, highlighting how expert activations contribute to the classification output.

## 3.1 ViT-based Temporal Generation.

The first component of our framework is a ViT (Dosovitskiy et al., 2021) adapted for temporal sequence generation. We represent the multi-band EEG data as an image tensor $X \in \mathbb{R}^{B \times C \times T_{\text{in}}}$, where the three dimensions correspond to sub-bands ($B = 3$), channels ($C$), and time samples ($T_{\text{in}}$). Following the ViT formulation, the signal is partitioned into non-overlapping patches of size $(B, C, t)$, where $t$ denotes a small temporal window (e.g., $t = 10$ samples $\approx 0.04\,\text{s}$). Each patch is then flattened and linearly projected into a latent embedding, forming a sequence of tokens. Like standard ViTs, we include positional embeddings added to the patchified embedding. A learnable class token is attached to the sequence, which serves to expand the representation to match the target output length $T_{\text{out}}$. This latent sequence is then processed by a transformer-based decoder (also implemented as a ViT), and the resulting patches are unpatchified to reconstruct the output sequence $\hat{X} \in \mathbb{R}^{B \times C \times T_{\text{out}}}$, with $T_{\text{out}} > T_{\text{in}}$. The model was optimized using the mean squared error (MSE) loss. We clip the extended sequence of shape $\mathbb{R}^{B \times C \times (T_{\text{out}} - T_{\text{in}})}$, which represents the newly generated temporal samples. This generated segment was concatenated with the original input $X$, forming the final output representation.

## 3.2 MoE with Channel-Subband Temporal Decoder

The second part of our framework is a decoder that jointly models subband, channel, and temporal dependencies using a MoE design.

### 3.2.1 MoE

MoE (Shazeer et al., 2017) mechanism is designed to increase model capacity while keeping computational cost manageable through sparse activation. MoE has also been successfully applied to EEG decoding tasks (Yang et al., 2025). Instead of applying a single shared transformation to all inputs, an MoE layer maintains a set of $E$ experts $\{f_1, f_2, \ldots, f_E\}$, each parameterized as a learnable function (e.g., convolutional filters in our case). For each input token $x$, a gating network produces a probability distribution over experts, and only the top-$k$ experts are selected to process the input. Specifically, we take the mean over temporal domain as the input of the gating network, where the network consist of two layers of MLP and one ReLU as activation. The final output is then obtained as a weighted combination of the selected experts' outputs, where the gating scores act as mixture coefficients. This strategy enables different experts to specialize on distinct temporal or spectral patterns in the EEG signal, enhancing both representation power and generalization.

**Auxiliary Load-Balancing Loss.** To encourage balanced utilization of experts and prevent overfitting, we introduce an auxiliary load-balancing loss. For each MoE layer, the gating network computes a probability distribution over $E$ experts for each input token. Let $G \in \mathbb{R}^{B \times E}$ denote the gate probabilities for a batch of $B$ inputs, with $G_{ij}$ representing the probability of assigning input $i$ to expert $j$. The mean usage of each expert is then $\bar{u}_j = \frac{1}{B} \sum_{i=1}^{B} G_{ij}$. We define the auxiliary loss as the Kullback-Leibler (KL) divergence between the mean expert usage and a uniform distribution:

$$\mathcal{L}_{\text{aux}} = \lambda_{\text{aux}} \, \text{KL}\Big( \log(\bar{u}) \, \| \, U \Big)$$

where $U$ is a uniform vector of length $E$, and $\lambda_{\text{aux}}$ is a weighting coefficient. This loss encourages all experts to be used approximately equally, preventing collapse onto a small subset of experts and improving generalization.

### 3.2.2 Channel-Subband fused Temporal Decoder

The decoder first integrates information across sub-bands and channels by treating the data as a combined subband-channel dimension of shape $(B * C, 1, T)$. A convolutional layer with kernel $(1, 1)$ is applied to extract higher-level spectral-channel features. Next, a subband-channel-temporal block is applied that consists of two convolutional layers: one over the subband-channel dimension and one over the temporal dimension. This block pools information separately from the spectral-channel and temporal dimensions. The block is repeated twice to progressively capture richer patterns across

channels, sub-bands, and time. Nonlinearities and dropout are applied between layers for regularization. Finally, the extracted features are flattened and passed through a fully connected layer to produce the class logits. In our model, we replace the last temporal convolution layer with a MoE layer, where experts in the networks employ the original temporal conv.

## 3.3 Data Augmentation

In the following section, the dataset is denoted as $X \in \mathbb{R}^{S \times B \times N \times M \times T \times C}$, where $S$ is the number of subjects, $B$ the number of subbands, $N$ the number of trials, $M$ the number of targets, $T$ the temporal length, and $C$ the number of EEG channels.

**Cross-Subject Temporal Stitching**  As a generalization of (Lotte, 2015), the EEG signals are first divided into short temporal segments, or *chunks*, across the time dimension. For each chunk, we randomly select a segment of the same duration from any subject and trial in the dataset, preserving the original subband, channel, and target labels. The selected segments are combined across time to form a new synthetic trial, maintaining the original subband, channel, and target structure. This approach allows the creation of entirely new temporal patterns by sampling from different subjects and trials, rather than modifying the original trial. Namely, $\oplus_i X_{s_i,:,n_i,:,i\tau:\tau(i+1),:} \in \mathbb{R}^{1 \times B \times 1 \times M \times T \times C}$ is a generated piece of data of, where $\{0, \tau, 2\tau, ...\}$ is the time chunk sequence and each $s_i, n_i$ is randomly selected among $S, N$,

**Channel Chunk Shuffle**  Given a random subject, input data is first divided into consecutive chunks along the time dimension. For each chunk, with a certain probability, two channels are randomly selected and swapped, while all other dimensions—including subbands, trial, target labels remain unchanged. Explicitly, given a subject's trial and target,if time chunks $(t_i\tau, t_j\tau)$ are selected to shuffle with $(c_{i_0}, c_{i_1}), (c_{j_0}, c_{j_1}) \in S_c$ are transpositions corresponding to $t_i, t_j$ as channel swap, $X_{s,:,n,m,t_i\tau:t_{i+1}\tau,c_{i_0}}$ replaces the channel $c_{i_1}$ and similarly for $t_j, (c_{j_0}, c_{j_1})$.

**Random Temporal Crop**  Gaze-shift and stimulus-locking latencies can vary across individuals in practical systems. Such variability has been extensively documented in the SSVEP literature (Liu et al., 2021b; Wang et al., 2016; Pan et al., 2011; Lemm et al., 2005; Dornhege et al., 2006; Wu et al., 2008; Qi et al., 2015) Therefore, inspired by (Liu et al., 2021b), a *Random Temporal Crop (RTC).* augmentation is utilized to increase temporal diversity in the training data. For any chosen subject, we preserve the original trial, target label, subband, and channel structure.With a given probability, we randomly select a short segment of time (e.g. 0.03s) from the data, keeping only the latter portion and discarding the former. The cropped segment is zero-padded at the end to restore the trial to its original temporal length.

**Channel Decorrelation**  We adopt a covariance-based whitening procedure across channels, conditioned on each subject and subband. For each subject and subband, we first compute the mean trial across all training trials to obtain a representation. This mean trial is used to estimate the channel covariance, from which a whitening matrix is derived (He & Wu, 2019; Liu et al., 2021a). The whitening matrix is applied to both training and test data, effectively reducing subject, trial-level variability while preserving the temporal and target-related structure of the signals. The decorrelation procedure emphasizes stable patterns across different subject and subband.

## 3.4 Transfer Learning.

Following the transfer learning strategy of (Guney et al., 2022), we adopt a staged training procedure to strengthen representation ability. Our model comprises two main components: a ViT encoder for temporal length generation and a MoE-based decoder for subband-channel and temporal integration. Thus, the transfer learning process is added to these component.

In the first stage, the ViT encoder is trained in a generative manner, reconstructing the temporal sequence from shorter inputs. In the second stage, this encoder is fine-tuned separately for each subject, where the global model parameters are re-initialized and adapted using only subject-specific data. The decoder is trained in a similar two-step fashion: first, a global MoE decoder is optimized using the pooled training data across all subjects, and subsequently, a subject-specific fine-tuning step is applied to adapt the decoder to individual variability.

# 4 EXPERIMENTS

## 4.1 DATASET

The experiments were carried out on two public 40-target SSVEP datasets: the Benchmark dataset (Wang et al., 2016) and the BETA dataset (Liu et al., 2020). Both datasets employed the joint frequency and phase modulation (JFPM) method to encode target stimuli. The data acquisition equipment for the Benchmark and BETA datasets is identical; however, the Benchmark dataset was collected in a controlled laboratory environment within an electromagnetic shielding room, whereas the BETA dataset was recorded in a more naturalistic setting, reflecting real-world conditions. All experiments, including comparisons with state-of-the-art methods, were performed on these two datasets. This allows us to evaluate the performance of our decoding approach under both controlled and realistic acquisition conditions.

## 4.2 PREPROCESSING

The same preprocessing pipeline was applied to both datasets. Nine electrodes (Pz, PO5, PO3, POz, PO4, PO6, O1, Oz, and O2) were selected for analysis. The EEG signals were downsampled to 250 Hz. To account for visual response latency, we considered delays of $0.14$ s for Benchmark and $0.13s$ for BETA, consistent with previous studies (Chen et al., 2015b). For each trial, data segments of length $t$ seconds were extracted in the time windows $[0.14, 0.14 + t]$ s and $[0.13, 0.13 + t]$ s after stimulus onset for Benchmark and BETA, respectively.

We apply a filter-bank approach as a preprocessing step to enhance SSVEP signals (Chen et al., 2015a). Data passes through three band-pass filters with frequency ranges $(8N, 90)$ Hz, where $N = 1, 2, 3$, and filtered signals are concatenated along the sub-band dimension. This procedure captures multiple harmonics and improves the signal representation for subsequent decoding.

## 4.3 BASELINE MODELS

**Deep Learning Models.** **DNN** (Guney et al., 2022) is a dense convolutional neural network that processes time-series data and incorporates a fine-tuning stage to boost performance. **SSVEPformer** (Chen et al., 2023) is a transformer-based neural network that takes complex spectra as input, leveraging a transformer encoder and fully connected layer to extract phase and frequency features. **TR-CANet** (Deng et al., 2023) applies TRCA-based spatial filtering to the input data, followed by a DNN for feature learning.

**Traditional Models.** **TDCA** (Liu et al., 2021b) addresses the redundancy of stimulus-specific spatial filters in TRCA and the underutilization of temporal information. It enhances the performance of individually calibrated SSVEP-BCIs by learning task-discriminative spatiotemporal components. **TRCA** (Nakanishi et al., 2017) derives spatial filters by maximizing SSVEP reproducibility across trials, while eTRCA extends this by ensembling filters across all frequencies. **eCCA** (Nakanishi et al., 2014) introduces a combination of spatial filters derived from canonical correlation analysis (CCA) and employs a user-specific target identification algorithm based on individual calibration data. **msTRCA** (Wong et al., 2020) extends TRCA with a multi-stimulus learning scheme that leverages data from both target and non-target stimuli.

## 4.4 EXPERIMENTAL SETUP

We employed $k$-fold cross-validation, with $k = 6$ for Benchmark and $k = 4$ for BETA. For each subject, one block of EEG data was designated as the test set, while the remaining blocks were used for training within that fold. All training follows the four-stage procedure: ViT generative pretraining, ViT subject-specific fine-tuning, MoE decoder pretraining, and MoE subject-specific fine-tuning. Further implementation details are provided in the relevant subsectionA of the Appendix.

## 5 RESULT

To evaluate the performance of algorithms among different data lengths, we report both classification accuracy and ITR. The ITR, measured in bits per minute (bpm), is defined as (Wolpaw et al., 2002):

$$\text{ITR}(P, T, M) = \left( \log_2 M + P \log_2 P + (1 - P) \log_2 \frac{1 - P}{M - 1} \right) \frac{60}{T} \tag{1}$$

Here, $M$ denotes the number of target classes, $P$ denotes the classification accuracy, and $T$ (in seconds) represents the total selection duration, including gaze time and a fixed gaze shift of 0.5 s.

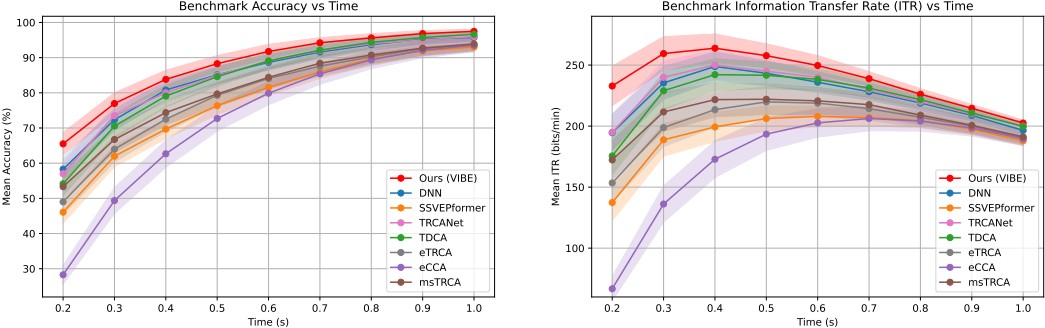

Figure 2: The left panel shows the mean classification accuracy, and the right panel shows the mean information transfer rate (ITR) across all 35 subjects in the Benchmark dataset. Shaded regions indicate the standard errors for subjects.

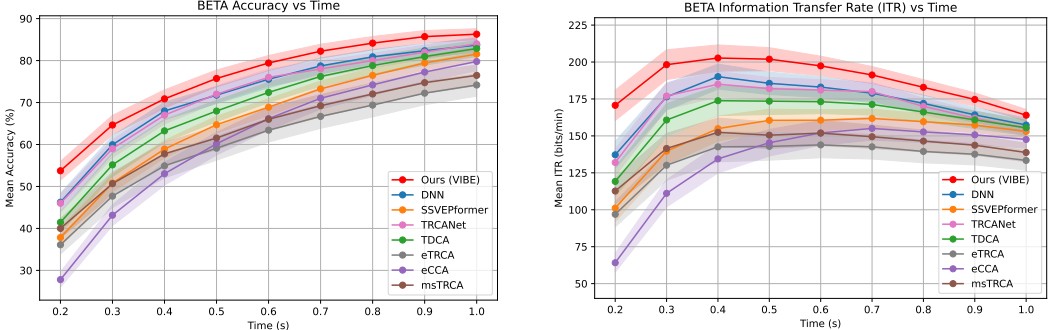

Figure 3: The left panel shows the mean classification accuracy, and the right panel shows the mean ITR across all 70 subjects in the BETA dataset. Shaded regions indicate the standard errors for subjects.

Figures 2 and 3 present the average classification accuracy and ITR of the proposed VIBE network evaluated on Benchmark and BETA across different data lengths. At the shortest data length (0.2 s), VIBE achieved the largest accuracy advantage over all other methods, highlighting its superior capability for rapid SSVEP decoding (Benchmark: 65.5% vs. 58.8%; BETA: 54.1% vs. 46.2%). The maximum ITR for VIBE was observed at 0.4 s, reaching $263.8 \pm 11.7$ bpm for Benchmark and $202.7 \pm 8.9$ bpm for BETA, exceeding the corresponding values of the DNN baseline ($248.8 \pm 11.8$ bpm and $190.1 \pm 8.4$ bpm, respectively). For a 1 s data length, VIBE maintained superior classification performance compared with DNN (Benchmark: $97.4 \pm 0.7\%$ vs. $95.7 \pm 1.1\%$; BETA: $86.3 \pm 1.3\%$ vs. $83.7 \pm 1.6\%$). Collectively, these results demonstrated that VIBE effectively decodes SSVEP responses across a range of time windows, with particularly pronounced benefits under short observation periods.

The performance of each method was evaluated in terms of decoding accuracy and ITR across data lengths. A two-way (method × data length) repeated-measures ANOVA (Greenhouse–Geisser

corrected) revealed a statistically significant interaction between method and data length for both datasets (Benchmark: $F(56, 1904)$, $p < 0.001$; BETA: $F(56, 3864)$, $p < 0.001$). The eight methods included in this analysis correspond to those described in the Experiments Section. The detailed results are provided in Appendix Table 7. These findings indicate that the effect of data length on decoding performance depended on the method used, and vice versa, highlighting significant differences in performance trends across methods and data lengths.

For both Benchmark and BETA, paired t-tests revealed that our proposed VIBE method achieved significantly higher decoding accuracies than the deep learning baseline (DNN) and the traditional method (TDCA) across all evaluated data lengths (all: $p < 0.05$). The details of these results are summarized in Table 8, in Appendix. The advantage of VIBE was especially pronounced at short data lengths (e.g., 0.2 s, Benchmark: VIBE vs. DNN: $p = 1.0 \times 10^{-12}$; VIBE vs. TDCA: $p = 2.2 \times 10^{-14}$; BETA: VIBE vs. DNN: $p = 4.8 \times 10^{-18}$; VIBE vs. TDCA: $p = 3.3 \times 10^{-25}$), demonstrating that our method was more robust under very short EEG segments. As data length increases, all methods converged towards similar performance, but VIBE consistently maintained a significant edge, indicating its effectiveness in both short- and long-window SSVEP decoding. After applying the Holm-Bonferroni correction to control for multiple comparisons, all paired t-test results remained statistically significant (p < 0.05), indicating that VIBE consistently outperforms the compared methods across data lengths and datasets.

## 5.1 Ablation Study

To better understand the contribution of each component in our framework, we conducted an ablation study on the Benchmark dataset with a 0.2 s data length. A brief summary of the ablation is presented in Table 10. The original model achieved an accuracy of **65.5%**.

**MoE** Removing the MoE module resulted in an accuracy of **64.2%**, highlighting its importance. Further analysis of MoE placement across different layers is provided in Appendix D.1.

**ViT regeneration.** Removing the ViT regeneration module led to a performance drop to **61.8%**, highlighting its essential role in feature representation. Further exploration of the effect of varying ViT generation time length is provided in Appendix D.3, which indicated that the optimal generation time depended on the input trial length: longer generation times benefited short trials, while shorter generation times were preferable for longer trials.

**Data augmentation.** Two augmentation strategies were employed: a decorrelation-based augmentation and an additional data generation module (the three methods described in Section 3.3) for the MoE decoder. When only the decoder-specific augmentation was removed, the accuracy decreased to **63.5%**; when both strategies were removed, the accuracy further dropped to **62.1%**. Further analysis of the effect of removing each augmentation is provided in Appendix D.2.

These results confirm that each module contributes positively, with all these three blocks play critical roles, and that the optimal ViT generation time length is data-length dependent.

## 5.2 Feature Visualization via t-SNE

To explore the reasons behind the superior performance of our model, we applied t-distributed Stochastic Neighbor Embedding (t-SNE) (Maaten & Hinton, 2008) to visualize the features learned from the final fully connected layer. We examined only our model, comparing the full version with an ablated version in which the ViT, data augmentation, and MoE were removed, using a data length of 0.6 s. In Appendix Figure 4, dots of the same color in the full model (left) formed more compact and dense clusters than in the ablated model (right). The circled clusters highlighted representative examples. This increased density indicated that the three key modules contributed to generating more discriminative and tightly grouped feature representations.

## 5.3 Subject-wise ITR Visualization

In Appendix Figure 5, we visualized the ITR for each subject using radar plots to compare our model (VIBE) with baseline methods (DNN and TDCA) at a data length of 0.2 s. Four radar plots were presented, corresponding to the Benchmark and BETA datasets, and comparing VIBE with DNN and TDCA, respectively. Each spoke in the radar plot represented an individual subject, and the distance

from the center indicated the ITR value. Across both datasets, VIBE consistently achieved higher ITR values for most subjects compared to the baseline models, illustrating its superior performance and robustness in short-duration SSVEP decoding.

# 6 DISCUSSION

## 6.1 NEURAL UNDERPINNINGS OF THE PROPOSED MODULES

In our study, the effectiveness of ViT-based regeneration can be attributed to the temporal nature of the SSVEP signals. Thus, the regenerated segments effectively extended the temporal window available to the decoder, which provided richer frequency-level information (target frequencies lie within the 8-15.8 Hz range). The regeneration step ensured that the decoder could access more complete frequency cycles, especially when the original data length was short. Although each ViT patch embedding only encoded a tiny fraction of data length (e.g. 0.04 s), it preserved additional temporal information for the decoder to facilitate classification.

Data augmentation plays a crucial role in enhancing the robustness of the model by introducing variability and simulating real-world scenarios. Several techniques have been implemented in this study, each inspired by physiological and contextual considerations related to EEG signals. First, the random temporal crop augmentation addresses inter-subject and task-dependent variability in SSVEP latency. While the standard latency is incorporated in the data preprocessing pipeline, the actual latency for each individual can differ, so this augmentation randomly samples temporal segments within each trial to learn latency-tolerant features rather than overfitting to a fixed window, improving generalization. Second, the channel chunk shuffle augmentation is motivated by the dipole-source origin of EEG and distortions from volume conduction and other artifacts, and it randomly shuffles chunks of channels to simulate varied electrode placements and signal quality. This promotes invariance to sensor positioning and improves generalization across hardware setups and individuals. Third, cross-subject temporal stitching encourages the decoder to focus on frequency-level information rather than subject-specific features by stitching trials across subjects, exposing it to diverse temporal patterns and yielding generalized frequency responses that reflect underlying physiology. This increased diversity of neural signals mitigates overfitting to individual trials and promotes more robust class-specific representations, ultimately improving the model's generalization performance.

The MoE mechanism is particularly valuable in the final temporal convolution layer of the model, where different experts can specialize in learning distinct temporal patterns relevant to specific task targets. Some experts may focus on shorter, rapid temporal responses, while others may specialize in longer, more sustained patterns, enabling the model to better capture the full range of temporal dynamics. This adaptability allows the model to allocate different experts to process different parts of the temporal signal.

## 6.2 TRAINING AND TESTING TIME ANALYSIS

For VIBE, the two pretraining stages were performed using data from all subjects (excluding test data), while the fine-tuning stages employed data from a single subject. Table 9 in the Appendix summarizes the training times for each stage and the testing time for a single 0.4 s trial, with all experiments conducted on an NVIDIA RTX 4090 GPU. The pretraining stages accounted for the majority of the training time, whereas fine-tuning for a specific subject could be completed in approximately 17 seconds for BETA and 1 minute for Benchmark. The difference in training time between the two datasets is due to the differing number of epochs in each stage. Testing a single trial required less than 1 ms, which is negligible compared to the data duration. These findings indicate that VIBE provides a practical and efficient solution for SSVEP decoding in BCI applications.

## 6.3 LIMITATION AND FUTURE DIRECTION

One limitation of this work is that certain subjects exhibit performance that deviates markedly from the overall distribution, underscoring the need for more generalized approaches capable of handling inter-subject variability. Future investigations could therefore benefit from conducting experiments in alternative evaluation settings, such as performing cross-validation across subjects rather than

individually calibrated scenario, or evaluate on other EEG decoding tasks (Song et al., 2024; Jiang et al., 2024; Wang et al., 2023), to provide a more rigorous assessment of generalization. Finally, an important direction for future research is the implementation of online experiments, wherein new patients are directly evaluated, to provide a realistic assessment of the model's effectiveness in practical BCI applications.

## 7 CONCLUSION

In this study, we proposed the Vision Transformer Based Expert network (VIBE), a multistage deep learning framework that integrates a ViT-MoE architecture and novel data enhancement approaches tailored for SSVEP data. By leveraging information from short-duration EEG recordings, VIBE learns effective and discriminative neurophysiological representations for individually calibrated SSVEP decoding. Evaluations on two benchmark datasets demonstrate that VIBE significantly improves both decoding accuracy and ITR. Overall, these results establish VIBE as a strong candidate for SSVEP decoding and support continued progress in high-speed BCI research.

## 8 REPRODUCIBILITY STATEMENT

All datasets used in this work are publicly available and open-sourced. To facilitate reproducibility, we provide the complete code for our models and experiments alongside the submission in the supplementary material. Detailed descriptions of model architectures, training procedures, and data preprocessing steps are included in the main text, Appendix, ensuring that independent researchers can replicate our results.

## 9 ETHICS STATEMENT

This work adheres to the ICLR Code of Ethics. All datasets used are publicly available and open-sourced. Specifically, the Benchmark and BETA datasets were collected under protocols approved by the respective institutions; For example, the BETA dataset protocol was approved by the Ethics Committee of Tsinghua University (No. 20190002) as reported in the original publication. No additional human subjects were involved in this study. The study focuses on computational modeling and analysis, without potential for harmful applications. All authors have read and complied with the ICLR Code of Ethics.

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

## A    IMPLEMENTATION DETAILS

For both ViT stages, the patch temporal length is set to 10 and the hidden dimension to 48, with a dropout rate of 0.8. For all trial durations except 0.2 s, the generative time length is 0.04 s, while for trials of 0.2 s, the generative time length equals 0.2 s. In the MoE decoder stage, the MoE is applied only to the second temporal layer, using 4 experts. The gating network consists of two MLPs, each with an intermediate dimension of 100, and only the top expert is selected. For implementation convenience, the input data is reshaped to $(B \times C, 1, T)$, where $B$ represents the subband dimension. Consequently, all subband-channel layers use a kernel size of $(1, 1)$, and the two temporal convolution layers use a kernel size of 10. The output channels are 200 for all subband-channel layers and 120 for temporal layers. All dropout layers have a probability of 0.1, except for the layer before the final flattening and MLP, which uses 0.95. During fine-tuning, the dropout probability of all intermediate layers is reset to 0.5.

For data augmentation, cross-subject temporal stitching is performed with a time chunk of 30. For channel chunk shuffling, the chunk size is 20 with a swap probability of 0.3. Random temporal cropping is applied with an activation probability of 0.4, selecting a short segment of 0.02 s to 0.06 s. For each augmentation method, additional data corresponding to 20% of the original dataset size is generated.

The ViT learning rate is set to 0.0001 during general pretraining and 0.00001 during subject-specific fine-tuning, while the decoder learning rate is fixed at 0.0001. The Adam optimizer is used with a weight decay of 0.0001, and an L2 regularization penalty of 0.001 is applied to the decoder. The batch size for both datasets is 32.

The number of training epochs for each stage differs between the Benchmark and BETA datasets, as summarized in Table 1.

Table 1: Stage-wise training epochs for Benchmark and BETA datasets.

| Dataset | ViT Pretrain | ViT Transfer | Decoder Pretrain | Decoder Transfer |
|---------|--------------|--------------|------------------|------------------|
| Benchmark | 300 | 1000 | 1500 | 1000 |
| BETA | 300 | 500 | 500 | 700 |

# B MODEL SIZE COMPARISON

We report both the total number of trainable parameters and the corresponding memory usage in FP32 precision. As summarized in Table 2, all models remain lightweight, with memory requirements well within the range suitable for real time or embedded deployment. For this analysis, we only consider the deep learning–based models. Traditional methods are excluded because they do not maintain persistent trainable parameters and their memory footprint is dominated by temporary data buffers rather than model weights, making a direct comparison of "model size" with neural architectures not meaningful. Note that TRCANet and the DNN share the same decoder architecture, resulting in exactly the same number of parameters. This comparison highlights the balance between model capacity and efficiency across the evaluated architectures.

Table 2: Model size comparison: parameters and approximate memory (FP32).

| Method | Parameters | Approx. Size (MB, FP32) |
|--------|-----------|-------------------------|
| DNN | 0.18M | 0.7 |
| TRCANet | 0.18M | 0.7 |
| SSVEPformer | 9.26 M | 37.1 |
| Ours | 1.18M | 4.7 |

# C DECORRELATION DETAILS

For each subject $s$ and subband $b$, we first compute the mean across training trials:

$$\mu^{(s,b)} = \frac{1}{N} \sum_{n \in \text{train trials}} X_{n,:,:,:}^{(s,b)} \in \mathbb{R}^{M \times T \times C}. \tag{2}$$

The aggregated mean activity is reshaped into $\mu^{(s,b)} \in \mathbb{R}^{C \times (M \cdot T)}$, and used to compute the channel covariance matrix:

$$\text{Cov}^{(s,b)} = \frac{1}{M \cdot T} \mu^{(s,b)} \left( \mu^{(s,b)} \right)^{\top} \in \mathbb{R}^{C \times C}. \tag{3}$$

The whitening matrix is defined as

$$W^{(s,b)} = \left( \text{Cov}^{(s,b)} \right)^{-\frac{1}{2}}, \tag{4}$$

and decorrelation is applied to both training and test data as

$$\tilde{X}_{n,:,:,:}^{(s,b)} = W^{(s,b)} X_{n,:,:,:}^{(s,b)}, \quad \forall n. \tag{5}$$

By using the trial-averaged activity to construct the covariance, this procedure reduces trial-level variability while preserving target and temporal structure, and ensures that whitening is guided by stable patterns rather than noisy single-trial fluctuations.

# D FURTHER ABLATION STUDY

## D.1 MOE

The results of applying the MoE module at different layers are summarized in Table 3. When MoE was applied to the first subband–channel layer, the accuracy was **65.0%**, and applying it to the second subband–channel layer yielded **65.2%**. In contrast, applying MoE to the first temporal layer

resulted in a lower accuracy of **63.9%**, while placing it on the second temporal layer achieved an accuracy of **64.8%**. These results suggest that the second temporal layer and the subband layer were particularly important for MoE, as they contributed more significantly to improving performance compared to other layers. This highlights the importance of capturing frequency and temporal dynamics at these stages of the model.

Table 3: MoE Ablation Study: Different MoE Configurations. The Subbands listed refer to the Sub-band channel Conv layer in Subband-channel temporal blocks. Tested on Benchmark for 0.2s.

| MoE Type | Subband 1 | Subband 2 | Both Temporal | Temporal 1 |
|---|---|---|---|---|
| Accuracy (%) | 65.0 | 65.2 | 64.8 | 63.9 |

## D.2 DATA AUGMENTATION

The impact of different data augmentation strategies is summarized in Table 4. Removing temporal stitching, channel shuffle, or temporal crop resulted in minor decreases in accuracy of around 1%, while omitting decorrelation caused the largest drop to **63.3%**. These results indicate that all augmentation components contributed to model performance, with decorrelation having the most significant effect. Notably, removing all three data generation methods resulted in a 2% decrease, suggesting that each method provided complementary benefits along different dimensions. Table 5 further shows the effect of progressively adding augmentation components, with accuracy steadily increasing from using only decorrelation to including stitching, crop, and shuffle.

Table 4: Data Augmentation Ablation Study: Different Data Augmentation Configurations. Tested on Benchmark for 0.2s.

| Augmentation | No Stitching | No Channel Shuffle | No Temp Crop | No Decorrelation |
|---|---|---|---|---|
| Accuracy (%) | 64.5 | 64.9 | 64.7 | 63.3 |

Table 5: Data Augmentation Ablation Study: Different Data Augmentation Configurations, showing the effect of progressively adding augmentation components. Tested on Benchmark for 0.2s.

| Augmentation | Decorrelation | Decorrelation + Stitching | Decorrelation + Stitching + Crop | Decorrelation + Stitching + Crop + Shuffle |
|---|---|---|---|---|
| Accuracy (%) | 64.1 | 64.7 | 64.9 | 65.5 |

## D.3 EFFECT OF VIT GENERATION TIME LENGTH

Table 6 shows the impact of varying the ViT generation time length on classification accuracy. On the Benchmark dataset at 0.2 s, performance improved steadily from **64.5%** ($0.04s$) to **65.5%** (0.2 s). On the BETA dataset, a similar trend was observed, with accuracy increasing from **52.47%** (0.04 s) to **54.13%** (0.2 s). We note that for other data lengths (0.3 s to 1.0 s), the generated data augmentation achieving the best result was fixed at 0.04 s. To illustrate the effect of longer generation times, we performed the same experiment on the 0.4 s data length. The shortest generation time of 0.04 s achieved the highest accuracy (Benchmark: **83.85%**, BETA: **70.85%**), while increasing the generation time gradually decreased performance across other settings by up to 1.5%.

# E SUPPLEMENTARY TABLES AND FIGURES

Table 6: Effect of ViT generation time length on classification accuracy (%).

| Dataset | 0.04 s | 0.08 s | 0.12 s | 0.16 s | 0.20 s |
|---|---|---|---|---|---|
| Benchmark (0.2 s) | 64.50 | 64.91 | 65.28 | 65.01 | **65.50** |
| BETA (0.2s) | 52.47 | 52.64 | 52.72 | 53.26 | **54.13** |
| Benchmark (0.4 s) | **83.85** | 83.14 | 82.60 | 82.64 | 82.71 |
| BETA (0.4s) | **70.85** | 70.50 | 69.96 | 69.17 | 69.50 |

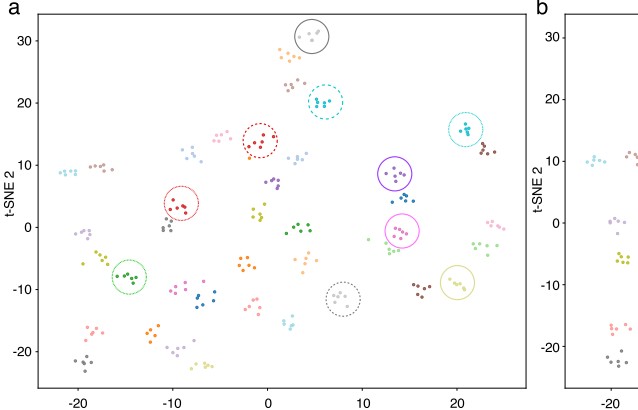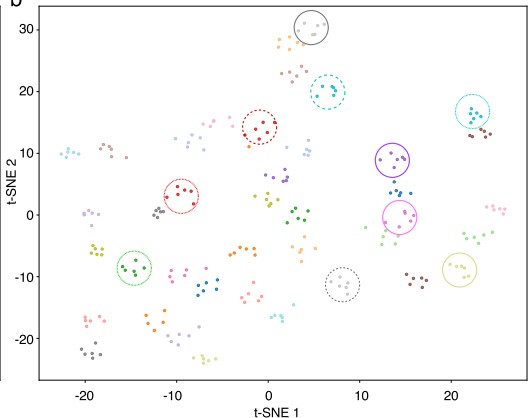

Figure 4: t-SNE visualization of learned features for a representative subject from the Benchmark dataset, using a data length of 0.6 s. Left: full model; Right: ablated model (without ViT, data augmentation, or MoE).

Table 7: Greenhouse–Geisser corrected two-way repeated-measures ANOVA results for the interaction effect between data length and method.

| Effect | Benchmark (Accuracy) | Benchmark (ITR) | BETA (Accuracy) | BETA (ITR) |
|---|---|---|---|---|
| $F$ | 22.598 | 22.336 | 25.256 | 32.494 |
| $p_{\text{GG}}$ | $3.45 \times 10^{-7}$ | $6.17 \times 10^{-7}$ | $7.05 \times 10^{-8}$ | $4.86 \times 10^{-10}$ |

Table 8: Paired t-test p-values comparing VIBE with DNN and TDCA for Benchmark and BETA across data lengths (0.2–1.0 s).

| Data length (s) | Benchmark | | BETA | |
|---|---|---|---|---|
| | VIBE vs DNN | VIBE vs TDCA | VIBE vs DNN | VIBE vs TDCA |
| 0.2 | $1.044 \times 10^{-12}$ | $2.211 \times 10^{-14}$ | $4.821 \times 10^{-18}$ | $3.305 \times 10^{-25}$ |
| 0.3 | $1.713 \times 10^{-11}$ | $1.947 \times 10^{-7}$ | $7.752 \times 10^{-13}$ | $1.880 \times 10^{-17}$ |
| 0.4 | $2.147 \times 10^{-7}$ | $5.304 \times 10^{-7}$ | $1.144 \times 10^{-7}$ | $8.812 \times 10^{-13}$ |
| 0.5 | $1.269 \times 10^{-6}$ | $1.073 \times 10^{-5}$ | $2.888 \times 10^{-9}$ | $1.010 \times 10^{-14}$ |
| 0.6 | $2.425 \times 10^{-7}$ | $4.578 \times 10^{-5}$ | $1.933 \times 10^{-10}$ | $1.638 \times 10^{-14}$ |
| 0.7 | $8.017 \times 10^{-5}$ | $5.435 \times 10^{-3}$ | $1.100 \times 10^{-8}$ | $7.185 \times 10^{-11}$ |
| 0.8 | $1.284 \times 10^{-3}$ | $8.630 \times 10^{-3}$ | $1.463 \times 10^{-9}$ | $1.445 \times 10^{-11}$ |
| 0.9 | $1.398 \times 10^{-3}$ | $2.856 \times 10^{-2}$ | $6.985 \times 10^{-10}$ | $7.206 \times 10^{-9}$ |
| 1.0 | $2.875 \times 10^{-3}$ | $3.825 \times 10^{-2}$ | $1.275 \times 10^{-6}$ | $4.638 \times 10^{-7}$ |

Table 9: Training and testing times for VIBE on two datasets for time data length 0.4s. Times are in seconds, except for the test stages, which are in milliseconds.

| Dataset | ViT | | | MoE Decoder | | |
|---|---|---|---|---|---|---|
| | Train (s) | Finetune (s) | Test (ms) | Train (s) | Finetune (s) | Test (ms) |
| Benchmark | 270.8 | 19.4 | 0.7 | 4180.3 | 43.2 | 0.09 |
| BETA | 328.9 | 5.7 | 0.7 | 1681.1 | 11.8 | 0.09 |

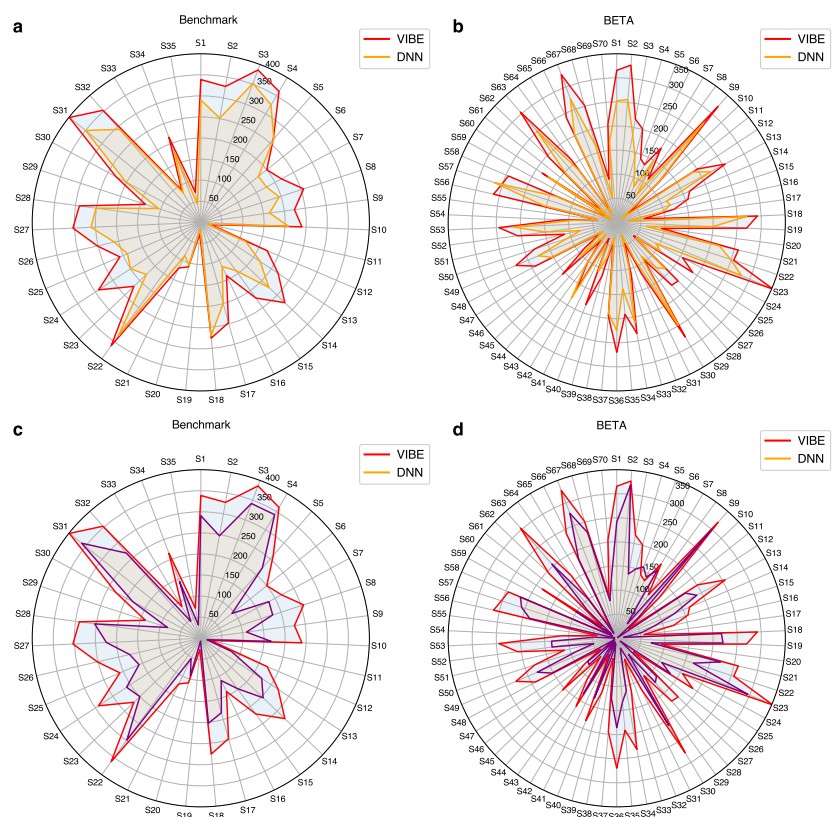

Figure 5: Subject-wise ITR plot. The methods were evaluated at a data length of 0.2s.

Table 10: Ablation General Results: No ViT, No MoE, and No Data Augmentation. Tested on Benchmark for 0.2s.

| Model | No ViT | No MoE | No Data Augmentation |
|---|---|---|---|
| Accuracy(%) | 61.8 | 64.2 | 62.1 |

## F LLM USAGE

Large language models (LLMs) were used solely for proofreading this manuscript to improve language clarity and readability. They were not involved in generating ideas, designing experiments, implementing methods, or analyzing results. All scientific contributions are entirely the work of the authors.

## G ANALYZING MULTI-STAGE CONTRIBUTIONS WITH MoE

In this analysis, we investigate how different stages of VIBE influence the model's frequency domain behavior under a controlled experimental setting. We use slightly modified parameters relative to the main experiments: each input consists of a 1s EEG segment, and the Stage 1 ViT generator produces an additional 0.2s extension. The decoder is implemented with eight temporal experts in the MoE layer. Both the Stage 3 (non-finetuned) and Stage 4 (finetuned) decoders are trained using the ViT augmented data, and all visualizations are performed using the same training set to remove cross-split variability. For each subject and trial, we extract the activation from the final temporal Conv2D layer and average across channels as well as subjects to obtain a single temporal sequence per target. This sequence is zero-padded to 5s to achieve sufficient spectral resolution for targets ranging from 8 Hz to 15.8 Hz, after which we apply an FFT. We compute the signal-to-noise ratio (SNR) by taking the peak amplitude at the target frequency and its second harmonic, normalizing each by the mean amplitude of neighboring frequency bins, and averaging the two values. We then perform paired t-tests to assess statistical significance. The analysis compares four conditions: (1) Stage 3 versus Stage 4 decoders, and (2) ViT augmented input (original+generated) versus original only input with zero-padding. The corresponding FFT visualizations and t-test statistics are summarized in the accompanying Figure 6 and Table 11. The FFT visualizations and t-test results consistently show that both the inclusion of ViT-generated data and decoder fine-tuning improve spectral responses. Specifically, the amplitude at the target frequency and its second harmonic is highest for FT with full input, followed by FT with original input, then BASE full, and finally BASE original. All comparisons are statistically significant, highlighting that ViT augmentation and Stage 4 fine-tuning are important for enhancing frequency-specific features in the model.

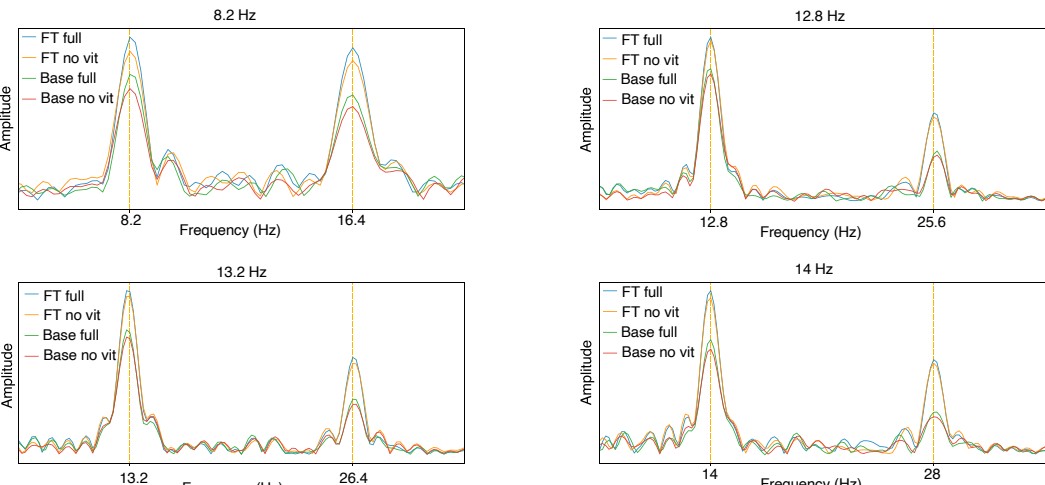

Figure 6: FFT visualizations of activation sequences from the final temporal layer. FT refers to Stage 4 and BASE refers to Stage 3. Full uses the complete input sequence (original data + Stage 2 ViT-generated data), while Original keeps only the original data and zero-pads the rest. Yellow dashed vertical lines indicate the target frequency and its second harmonic. The amplitude at the target frequency and its second harmonic is highest for FT with full input, followed by FT with original input, then BASE full, and finally BASE original. These results demonstrate that both incorporating ViT-generated data and fine-tuning the decoder strengthen the neuro physiological spectral responses.

Table 11: Paired t-test results comparing SNR differences from the final temporal layer activations. FT refers to Stage 4 and BASE refers to Stage 3. Full uses the complete input sequence (original data + Stage 2 ViT-generated data), while Original keeps only the original data and zero-pads the rest.

| Comparison | t-stat | p-value | n |
|---|---|---|---|
| FT (full − original) | 3.3941 | $1.59 \times 10^{-3}$ | 40 |
| BASE (full − original) | 3.0741 | $3.84 \times 10^{-3}$ | 40 |
| FT full − BASE full | 9.2720 | $2.08 \times 10^{-11}$ | 40 |
| FT original − BASE original | 7.6147 | $3.12 \times 10^{-9}$ | 40 |

## H  VISUALIZATION ANALYSIS FOR THE MoE TEMPORAL LAYER

We perform a qualitative analysis of the MoE temporal layer using the fine-tuned decoder for a randomly selected subject. Grad-CAM (Selvaraju et al., 2017) is applied to the final MoE temporal convolution layer to obtain activation importance over time. In Fig 8, the resulting temporal map is averaged across channels to produce a single 1D sequence. As in the previous analyses, this sequence is zero-padded to 5s to ensure sufficient spectral resolution before applying the FFT. In addition to the frequency-domain visualization, we record the MoE expert selection count Fig 7 for each target frequency to examine how the mixture-of-experts distributes attention across different spectral components. The MoE expert selection patterns reveal that the experts do not collapse into a single dominant expert; instead, different target frequencies elicit distinct expert activation distributions. This diversity indicates that individual experts specialize in different temporal–spectral patterns rather than redundantly modeling the same structure. Complementing this, the Grad-CAM analysis shows clear spectral peaks at the target frequency and its harmonic after FFT, demonstrating that the MoE temporal layer effectively pools and amplifies frequency-specific structure. Together, these results confirm that the MoE architecture meaningfully decomposes the temporal dynamics and contributes specialized processing across targets.

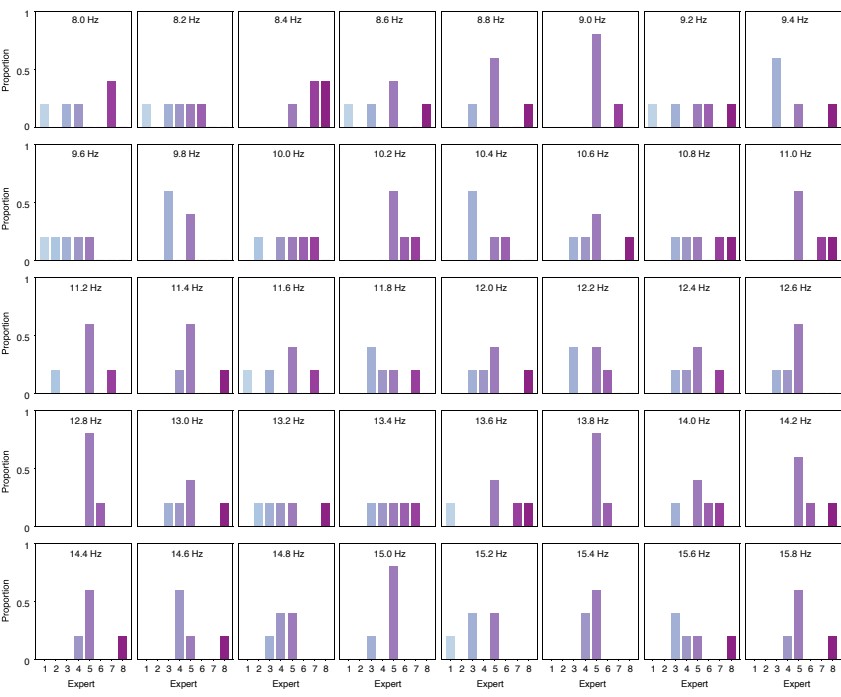

Figure 7: MoE expert selection counts for all 40 target labels in a randomly selected subject. The x-axis represents the different experts (1–8), and the y-axis shows the proportion of times each expert is selected. MoE expert activations vary systematically across target frequencies, reflecting frequency-specific patterns

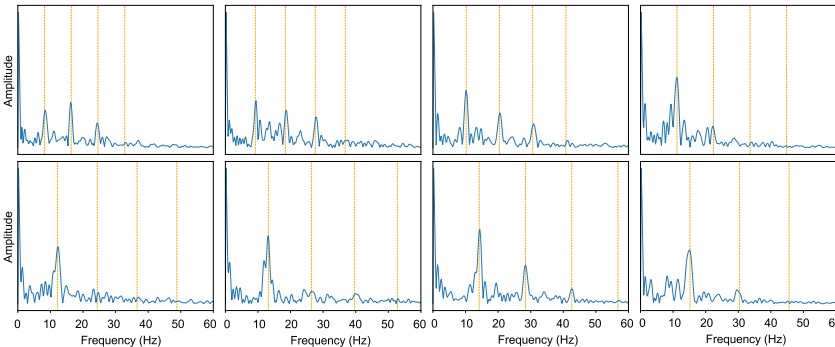

Figure 8: Grad-CAM visualization of the final MoE temporal layer, averaged across channels, for a randomly selected subject. FFTs are shown for selected target frequencies: 8, 9, 10, 11 Hz (top row) and 12, 13, 14, 15 Hz (bottom row) to illustrate temporal importance patterns. Yellow dashed vertical lines indicate the target frequency and its harmonics. Across all target frequencies, the Grad-CAM results consistently show peaks at the target frequency and its second harmonic, demonstrating that the MoE layer captures frequency-specific temporal activations.

## I SPECTRAL VALIDATION OF DATA AUGMENTATION

We provide spectral analysis for the two data augmentation methods used in our framework, illustrating their preservation on SSVEP frequency structure. Because stitch-augmented data and channel-chunk-shuffle augmentation are less intuitive than our other augmentation strategies, we include explicit spectral analyses to clarify how they operate.

### I.1 SPECTRAL VALIDATION OF STITCH-AUGMENTED DATA

To assess whether the Cross-Subject Temporal Stitching augmentation preserves useful SSVEP structure, we visualize the temporal and spectral profiles of the augmented samples.We use 1 s temporal length original data as input from Benchmark. After Stage 2 ViT generation (add 0.2s generate data) and Stitch recomposition, we select a single subband, average across channels and trials dimensions, zero-pad the resulting sequence to 5s, and apply an FFT. The corresponding temporal traces and spectra consistently exhibit clear peaks at the target frequency and its harmonic, indicating that the stitched signals retain the essential SSVEP signatures. Although Stitch mixes temporal chunks across channels and subjects, this variability diversifies the training distribution for the decoder while maintaining the frequency-specific structure required for accurate decoding. The accompanying plot (Fig. 9) contains two subpanels: the left panel shows the stitched time-series signal, and the right panel shows the FFT magnitude spectrum. In the spectral panel, clear peaks appear at the target frequency and its harmonic ($2\times$freq), confirming that the stitched samples preserve the characteristic SSVEP structure.

### I.2 SPECTRAL VALIDATION OF CHANNEL CHUNK SHUFFLE AUGMENTED DATA

To verify that the Channel Chunk Shuffle augmentation preserves essential SSVEP structure, we visualized the spectral profiles of augmented samples. We use 1 s temporal length original data as input from Benchmark. Following Stage 2 ViT generation (add 0.5s data) and shuffle recomposition, we select a single subband and channel, average across trials, zero-pad the resulting sequence to 5 s, and compute the FFT. By shuffling within defined temporal chunks, we expose the network to different channel arrangements, enhancing generalization while preserving physiologically meaningful spectral features. The accompanying plot (Fig. 10) shows that prominent peaks at the target frequency and its harmonic, indicating that the shuffled signals maintain the frequency-specific structure required for accurate decoding.

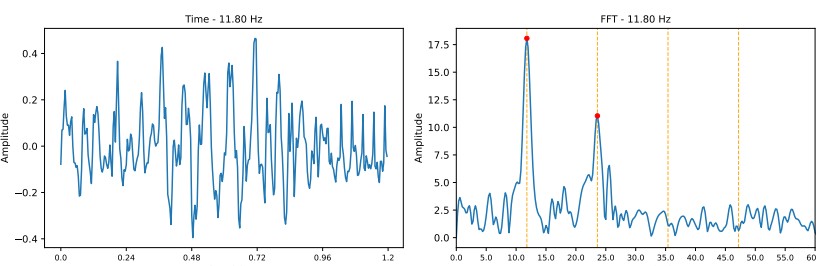

Figure 9: Time series (left) and corresponding FFT magnitude spectrum (right) of a Cross-Subject Temporal Stitching sample for a representative target frequency. We use 1 s temporal length original data as input from Benchmark and have ViT generate additional 0.2 s. In the spectral panel, the red dots on the curves indicate peaks at the target frequency and its second harmonic (2×freq), where target frequency and harmonics are highlighted by yellow dashed vertical lines. These clear spectral peaks confirm that the stitched samples preserve the characteristic SSVEP spectral structure.

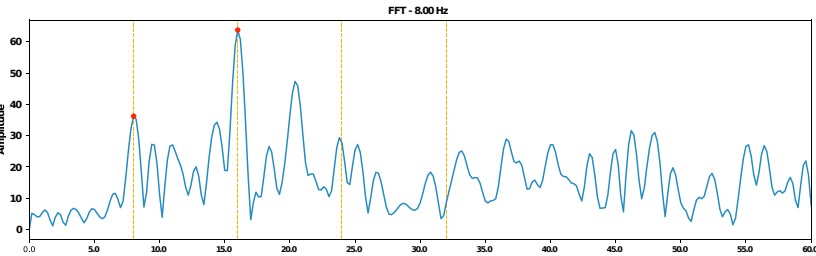

Figure 10: FFT magnitude spectrum (right) of a Channel Chunk Shuffle sample for a representative target frequency. We use 1 s temporal length original data as input from Benchmark and have ViT generate additional 0.5 s. The red dots on the curves indicate peaks at the target frequency and its second harmonic (2×freq), where target frequency and harmonics are highlighted by yellow dashed vertical lines. These clear spectral peaks confirm that the re-composited sample preserve the characteristic SSVEP spectral structure.

## J    COMPREHENSIVE RESULT TABLES AND ADDITIONAL BASELINES

Tables 12,13,14,15 summarize the performance of our model compared to baseline methods under different time lengths (0.2 to 1.0 s) on both the Benchmark and beta datasets. Tables 1 and 3 report classification accuracy (%), while Tables 2 and 4 present ITR. Across all conditions, our model consistently outperforms all baseline methods, demonstrating superior accuracy and efficiency. Additionally, we include comparisons with three recently proposed methods, Dis-ComNet (Li et al., 2025b), SESCNN (Jin et al., 2024), and ConsenNet (Zhang et al., 2024), which were not included in the result Fig 2,3. Our model achieves the highest accuracy and competitive iteration performance across all settings, highlighting its effectiveness and robustness.

Table 12: Classification accuracy (%) of various methods at different time lengths (0.2 - 1.0 s) on the Benchmark dataset. Values are reported as mean ± standard deviation.

| Method | 0.2 | 0.3 | 0.4 | 0.5 | 0.6 | 0.7 | 0.8 | 0.9 | 1.0 |
|---|---|---|---|---|---|---|---|---|---|
| DNN | 58.3 ± 3.2 | 72.3 ± 3.0 | 80.8 ± 2.7 | 85.1 ± 2.5 | 88.6 ± 2.2 | 91.6 ± 1.8 | 93.7 ± 1.6 | 95.2 ± 1.3 | 95.8 ± 1.1 |
| TDCA | 54.1 ± 3.4 | 70.5 ± 3.4 | 79.1 ± 3.1 | 84.6 ± 2.7 | 89.1 ± 2.3 | 92.2 ± 2.0 | 94.3 ± 1.6 | 95.7 ± 1.2 | 96.6 ± 1.0 |
| TRCA | 49.0 ± 3.6 | 64.0 ± 3.6 | 72.5 ± 3.5 | 79.3 ± 3.2 | 84.1 ± 3.0 | 87.6 ± 2.8 | 90.2 ± 2.5 | 92.5 ± 2.1 | 93.6 ± 1.9 |
| eCCA | 28.3 ± 2.7 | 49.4 ± 3.7 | 62.7 ± 3.9 | 72.7 ± 3.6 | 79.9 ± 3.3 | 85.4 ± 3.0 | 89.4 ± 2.6 | 92.1 ± 2.1 | 93.5 ± 1.8 |
| msTRCA | 53.3 ± 3.4 | 66.7 ± 3.6 | 74.3 ± 3.4 | 79.8 ± 3.2 | 84.4 ± 3.0 | 88.4 ± 2.7 | 90.7 ± 2.4 | 92.7 ± 2.0 | 93.9 ± 1.8 |
| SSVEPformer | 46.1 ± 3.1 | 62.0 ± 3.5 | 69.7 ± 3.3 | 76.3 ± 3.1 | 81.5 ± 3.0 | 86.0 ± 2.7 | 89.7 ± 2.4 | 91.9 ± 2.0 | 93.0 ± 1.8 |
| TRCANet | 58.4 ± 3.1 | 72.3 ± 3.0 | 80.8 ± 2.7 | 85.2 ± 2.5 | 88.8 ± 2.2 | 91.5 ± 1.8 | 93.7 ± 1.6 | 95.2 ± 1.3 | 95.5 ± 1.1 |
| Dis-ComNet | 54.2 ± 2,9 | 68.4 ± 3.6 | 76.0± 3.1 | 82.6 ± 3.0 | 86.3 ± 2.7 | 89.1 ± 2.8 | 92.2 ± 2.2 | 93. ± 1.9 | 95.6 ± 1.8 |
| SESCNN | 60.1 ± 3.2 | 73.5 ± 2.9 | 81.7 ± 3.3 | 85.6 ± 2.9 | 88.7 ± 2.6 | 91.2 ± 2.6 | 93.6 ± 2.7 | 94.9 ± 2.5 | 95.6 ± 2.7 |
| Ours | 65.5 ± 3.2 | 77.0 ± 2.9 | 83.8 ± 2.5 | 88.3 ± 2.2 | 91.7 ± 2.1 | 94.2 ± 1.9 | 95.6 ± 1.6 | 96.8 ± 1.1 | 97.4 ± 0.7 |

Table 13: ITR (bits/min) of various methods at different time lengths (0.2 - 1.0 s) on the Benchmark dataset. Values are reported as mean ± standard deviation.

| Method | 0.2 | 0.3 | 0.4 | 0.5 | 0.6 | 0.7 | 0.8 | 0.9 | 1.0 |
|---|---|---|---|---|---|---|---|---|---|
| DNN | 194.6 ± 14.8 | 235.4 ± 13.5 | 248.8 ± 11.8 | 243.3 ± 10.2 | 236.0 ± 8.5 | 228.1 ± 6.9 | 218.9 ± 5.8 | 208.7 ± 4.4 | 196.5 ± 3.8 |
| TDCA | 173.9 ± 15.5 | 227.5 ± 15.3 | 240.6 ± 13.0 | 240.5 ± 11.0 | 237.3 ± 8.8 | 230.3 ± 7.4 | 220.8 ± 5.8 | 210.0 ± 4.4 | 199.1 ± 3.4 |
| msTRCA | 151.7 ± 16.1 | 197.3 ± 15.6 | 212.0 ± 14.2 | 218.5 ± 12.3 | 217.7 ± 10.7 | 213.4 ± 9.5 | 206.3 ± 8.0 | 199.2 ± 6.6 | 189.5 ± 5.7 |
| eCCA | 65.2 ± 10.0 | 134.8 ± 14.9 | 171.4 ± 15.0 | 192.2 ± 13.4 | 201.6 ± 11.7 | 205.2 ± 10.1 | 203.3 ± 8.3 | 197.8 ± 6.7 | 188.9 ± 5.6 |
| msTRCA | 170.5 ± 16.0 | 210.1 ± 15.8 | 220.1 ± 14.0 | 220.6 ± 12.5 | 219.5 ± 10.9 | 216.7 ± 9.4 | 208.2 ± 7.9 | 199.9 ± 6.5 | 190.6 ± 5.6 |
| SSVEPformer | 137.4 ± 13.9 | 188.8 ± 15.3 | 199.3 ± 13.5 | 206.2 ± 11.9 | 208.0 ± 10.7 | 207.0 ± 9.2 | 204.4 ± 7.8 | 197.5 ± 6.4 | 187.9 ± 5.6 |
| TRCANet | 196.3 ± 14.8 | 235.5 ± 13.5 | 248.6 ± 11.8 | 243.2 ± 10.2 | 236.2 ± 8.5 | 226.9 ± 6.8 | 217.8 ± 5.8 | 207.6 ± 4.4 | 194.8 ± 3.7 |
| Dis-ComNet | 185.2 ± 15.1 | 225.6 ± 13.8 | 236.0 ± 13.5 | 234.3 ± 10.3 | 230.2 ± 9.8 | 220.9 ± 8.3 | 210.3 ± 7.4 | 205.7 ± 6.9 | 198.1 ± 5.4 |
| SESCNN | 204.5 ± 14.5 | 237.8 ± 15.9 | 253.0± 12.8 | 245.4 ± 11.4 | 236.6 ± 10.7 | 229.3 ± 9.5 | 212.2 ± 7.1 | 204.1 ± 7.4 | 189.9 ± 6.1 |
| Ours | 232.8 ± 15.7 | 259.4 ± 14.1 | 263.8 ± 11.7 | 257.7 ± 9.4 | 249.6 ± 8.9 | 238.8 ± 7.2 | 226.1 ± 6.2 | 214.5 ± 5.1 | 202.5 ± 4.2 |

Table 14: Classification accuracy (%) of various methods at different time lengths (0.2 - 1.0 s) on the BETA dataset. Values are reported as mean ± standard deviation. ConsenNet was evaluated under a slightly different setting: all but one subject were used as their training set. In their fine-tuning stage, they used the first three calibration blocks from the new subject as training and the remaining blocks as testing. Thus, for each test trial in the BETA dataset, all other trials from that subject, as well as all data from other subjects, were seen during training. They used more data for training compared to our setup, incorporating all other trials from the test subject as well as data from all remaining subjects.

| Method | 0.2 | 0.3 | 0.4 | 0.5 | 0.6 | 0.7 | 0.8 | 0.9 | 1.0 |
|---|---|---|---|---|---|---|---|---|---|
| DNN | 46.2 ± 2.1 | 59.9 ± 2.1 | 68.0 ± 2.1 | 71.8 ± 2.0 | 75.5 ± 2.0 | 78.7 ± 1.9 | 80.9 ± 1.7 | 82.3 ± 1.7 | 83.7 ± 1.6 |
| TDCA | 41.4 ± 2.4 | 55.2 ± 2.8 | 63.3 ± 2.7 | 68.0 ± 2.6 | 72.4 ± 2.4 | 76.2 ± 2.3 | 78.8 ± 2.1 | 80.9 ± 2.0 | 82.9 ± 1.9 |
| msTRCA | 36.1 ± 2.2 | 47.7 ± 2.7 | 54.9 ± 2.9 | 59.1 ± 2.9 | 63.4 ± 2.8 | 66.7 ± 2.9 | 69.4 ± 2.8 | 72.2 ± 2.8 | 74.2 ± 2.7 |
| eCCA | 27.8 ± 1.8 | 43.2 ± 2.5 | 53.0 ± 2.7 | 60.2 ± 2.7 | 66.2 ± 2.6 | 71.0 ± 2.6 | 74.2 ± 2.5 | 77.3 ± 2.4 | 79.8 ± 2.2 |
| msTRCA | 40.0 ± 2.3 | 50.7 ± 2.7 | 57.8 ± 2.7 | 61.5 ± 2.7 | 66.0 ± 2.7 | 69.2 ± 2.7 | 72.0 ± 2.6 | 74.7 ± 2.5 | 76.5 ± 2.5 |
| SSVEPformer | 37.8 ± 1.9 | 50.8 ± 2.3 | 58.9 ± 2.4 | 64.7 ± 2.4 | 68.9 ± 2.4 | 73.3 ± 2.3 | 76.5 ± 2.3 | 79.4 ± 2.2 | 81.5 ± 2.1 |
| TRCANet | 46.1 ± 2.1 | 60.0 ± 2.1 | 67.9 ± 2.1 | 71.7 ± 2.0 | 75.6 ± 2.0 | 78.9 ± 1.9 | 80.9 ± 1.7 | 82.2 ± 1.7 | 83.8 ± 1.7 |
| SESCNN | 46.9± 3.2 | 58.9 ± 3.8 | 68.1± 3.5 | 71.2 ± 3.9 | 75.9 ± 4.1 | 79.1 ± 3.4 | 81.1 ± 4.3 | 82.8 ± 3.9 | 84.1 ± 4.7 |
| ConsenNet | - | - | 67.9 ± 2.1 | - | 77.7 ± 2.2 | - | 83.6 ± 1.9 | - | - |
| Ours | 54.1 ± 2.3 | 64.6 ± 2.0 | 70.8 ± 1.9 | 75.7 ± 1.9 | 79.4 ± 1.8 | 82.2 ± 1.8 | 84.2 ± 1.7 | 85.7 ± 1.6 | 86.3 ± 1.4 |

## K    COMPARISON OF GENERATORS FOR TEMPORAL SEQUENCE EXTENSION

To rigorously evaluate the design choice of using a ViT for temporal sequence generation, we conducted a comparative study against an alternative transformer-based architecture, the Convolutional

Table 15: ITR (bits/min) of various methods at different time lengths (0.2 - 1.0 s) on the BETA dataset. Values are reported as mean ± standard deviation. ConsenNet was evaluated under a slightly different setting: all but one subject were used as their training set. In their fine-tuning stage, they used the first three calibration blocks from the new subject as training and the remaining blocks as testing. Thus, for each test trial in the BETA dataset, all other trials from that subject, as well as all data from other subjects, were seen during training. They used more data for training compared to our setup, incorporating all other trials from the test subject as well as data from all remaining subjects.

| Method | 0.2 | 0.3 | 0.4 | 0.5 | 0.6 | 0.7 | 0.8 | 0.9 | 1.0 |
|---|---|---|---|---|---|---|---|---|---|
| DNN | 137.2 ± 9.1 | 176.4 ± 9.0 | 190.1 ± 8.5 | 185.6 ± 7.6 | 183.0 ± 7.0 | 178.9 ± 6.3 | 172.0 ± 5.5 | 164.2 ± 5.0 | 157.4 ± 4.6 |
| TDCA | 118.3 ± 9.7 | 159.9 ± 11.1 | 173.0 ± 10.1 | 172.8 ± 9.1 | 172.4 ± 8.1 | 170.6 ± 7.3 | 165.5 ± 6.5 | 160.2 ± 5.9 | 154.8 ± 5.2 |
| msTRCA | 95.8 ± 8.5 | 129.4 ± 10.2 | 141.8 ± 10.1 | 142.1 ± 9.3 | 143.2 ± 8.7 | 141.8 ± 8.3 | 138.8 ± 7.7 | 136.9 ± 7.2 | 132.8 ± 6.6 |
| eCCA | 63.1 ± 6.4 | 110.2 ± 9.1 | 133.6 ± 9.5 | 144.5 ± 9.0 | 151.2 ± 8.3 | 154.3 ± 7.8 | 151.9 ± 7.0 | 150.0 ± 6.5 | 146.8 ± 5.8 |
| msTRCA | 111.7 ± 9.0 | 140.5 ± 10.2 | 151.6 ± 9.7 | 149.7 ± 8.9 | 151.2 ± 8.5 | 148.8 ± 7.9 | 145.8 ± 7.3 | 143.0 ± 6.8 | 138.1 ± 6.2 |
| SSVEPformer | 101.1 ± 7.4 | 139.7 ± 9.2 | 154.9 ± 8.9 | 160.5 ± 8.4 | 160.6 ± 7.9 | 161.8 ± 7.3 | 159.6 ± 6.6 | 157.3 ± 6.1 | 153.0 ± 5.6 |
| TRC | 137.6 ± 9.0 | 176.7 ± 9.0 | 189.9 ± 8.5 | 185.4 ± 7.6 | 183.2 ± 7.0 | 179.9 ± 6.4 | 171.9 ± 5.5 | 163.6 ± 5.0 | 157.7 ± 4.6 |
| SESCNN | 141.2 ± 9.5 | 178.5 ± 10.3 | 189.3 ± 9.2 | 186.4 ± 8.4 | 182.8 ± 7.2 | 179.1 ± 6.5 | 171.1 ± 5.9 | 165.8 ± 4.3 | 159.1 ± 4.8 |
| ConsenNet | - | - | 188.7 ± 9.0 | - | 191.4 ± 7.7 | - | 181.8 ± 6.2 | - | - |
| Ours | 173.4 ± 10.4 | 199.1 ± 10.1 | 202.7 ± 8.9 | 201.9 ± 7.3 | 197.4 ± 7.2 | 191.2 ± 6.8 | 182.9 ± 5.9 | 174.7 ± 4.5 | 164.0 ± 4.1 |

Vision Transformer (CvT) (Wu et al., 2021). Both models were tasked with generating extended temporal sequences from short EEG segments of 0.2 s, while keeping all modules in multistages identical. This controlled setup isolates the effect of the generator architecture on the overall decoding performance. Quantitative results, summarized in Table 16, demonstrate that ViT consistently achieves higher mean accuracy in two datasets. These findings substantiate the selection of ViT as the temporal generator in VIBE, confirming its advantage for extending short EEG segments while preserving fine-grained, frequency-specific information critical for high-precision SSVEP decoding.

Table 16: Comparison of ViT and CvT as temporal sequence generators. All other components in the VIBE architecture remain fixed. The results show that ViT consistently outperforms CvT in mean accuracy, supporting its selection as the temporal generator.

| Model | Benchmark 0.2s | Benchmark 0.4s | BETA 0.2s | BETA 0.4s |
|---|---|---|---|---|
| ViT | 65.5 | 83.8 | 54.1 | 70.8 |
| CvT | 62.1 | 82.5 | 51.0 | 69.1 |

## L ELECTRODE CONFIGURATION AND SENSITIVITY

The choice of EEG electrodes can significantly impact both the performance and practicality of BCI systems. Different electrodes capture overlapping but distinct spatial patterns of brain activity, and their number and placement can influence classification accuracy. To evaluate this, we tested our model using several commonly adopted electrode sets(Liu et al., 2021b):

- **Central occipital montage (Nch = 3):** Oz, O1, O2

- **Classical occipital montage (Nch = 9):** Pz, POz, PO3/4, PO5/6, Oz, O1/2

- **Occipital montage (Nch = 21):** Pz, P1/2, P3/4, P5/6, P7/8, POz, PO3/4, PO5/6, PO7/8, Oz, O1/2, CB1/2

- **Parietal-occipital montage (Nch = 30):** CPz, CP1/2, CP3/4, CP5/6, TP7/8, Pz, P1/2, P3/4, P5/6, P7/8, POz, PO3/4, PO5/6, PO7/8, Oz, O1/2, CB1/2

- **Full montage (Nch = 64):** all channels

Our results Tab 17 show that classification accuracy improves as the number of EEG channels increases from 3 to 21, reaching a peak of 73.0% with 21 channels. However, using all 64 channels does not further improve performance and may reduce system practicality. Considering the trade-off between accuracy and usability, we select a 9-channel configuration (classical occipital montage)

for the experiments in this study. In Table 18, we report the maximum achievable ITR for the five electrode channels selection configurations. Compared with TDCA, our method almost consistently attains higher ITR across all channel combinations.

Table 17: Datalength of 0.2s over two datasets, classification accuracy (%) and ITR for different numbers of EEG channels.

| Nch | Benchmark Acc (%) | Benchmark ITR (bpm) | BETA Acc (%) | BETA ITR (%) |
|---|---|---|---|---|
| 3 | 40.7 | 117.2 | 31.8 | 80.3 |
| 9 | 65.5 | 232.8 | 54.1 | 173.4 |
| 21 | 73.0 | 272.3 | 60.0 | 202.8 |
| 30 | 71.5 | 263.9 | 58.0 | 192.0 |
| 64 | 67.4 | 240.3 | 49.9 | 153.3 |

Table 18: Maximum ITR across all time lengths for different numbers of EEG channels on the Benchmark and BETA datasets. For each channel configuration, we report the maximum achievable ITR over all evaluated time windows. Results compare our VIBE model with the TDCA baseline.

| Nch | Benchmark | | BETA | |
|---|---|---|---|---|
| | VIBE | TDCA | VIBE | TDCA |
| 3 | 142.5 | 153.6 | 119.3 | 101.3 |
| 9 | 263.8 | 244.5 | 202.7 | 173.6 |
| 21 | 290.7 | 281.0 | 216.2 | 198.0 |
| 30 | 289.0 | 281.7 | 213.0 | 195.6 |
| 64 | 267.3 | 275.3 | 189.5 | 181.9 |

