# OpenReview forum: "VIBE: Vision transformer based experts network for SSVEP decoding"
_ICLR.cc/2026/Conference — Submitted to ICLR 2026_

### Official Review · Reviewer_ideL · 2025-10-28

**Soundness:** 2
**Presentation:** 3
**Contribution:** 2
**Rating:** 4
**Confidence:** 2

**Summary:**

This paper presents VIBE, a new deep learning model designed to improve the decoding of SSVEPs for BCI. While the results reported are strong, the paper has several significant weaknesses that make it difficult to judge the true novelty and contribution of the work. The core issues revolve around motivation, technical justification, and experimental rigor.

**Strengths:**

Refer to Questions

**Weaknesses:**

Refer to Questions

**Questions:**

The introduction claims that current methods struggle to "jointly exploit local inductive biases and global dependencies." This statement is too general and high-level. It doesn't clearly pinpoint the specific shortcomings of existing models. For example, what exact "local inductive biases" are current CNNs missing? How do existing Transformers fail to capture "global dependencies" in the specific context of EEG signals? A more concrete problem definition is needed.

Furthermore, the literature review fails to capture the most recent advancements in the field. The most recent related work cited is from 2022, which is a considerable gap in the fast-moving area of deep learning for BCIs. For a paper submitted to ICLR 2026, it is expected to engage with work from 2023 and 2025. The absence of these references raises a critical question: Are the authors solving a problem that has already been addressed by more recent models? Without comparing against these newer methods, it's impossible to know if VIBE is truly state-of-the-art or simply reinventing the wheel.

The paper combines several existing components，a Vision Transformer (ViT), a Mixture of Experts (MoE) layer, and data augmentation, but the novelty of this combination is not well-justified. The idea of using a ViT to generate or extend a temporal sequence is interesting. However, the paper does not sufficiently explain why this is a better approach than other sequence generation or feature enhancement techniques. It feels more like an application of an existing architecture ("let's try ViT on EEG") rather than a principled innovation. The ablation study shows it helps, but the underlying "why" remains unclear. The decision to replace the last temporal convolution layer with an MoE layer is a major architectural choice, but it is made without strong justification. The authors provide an ablation table in appendix Table 2, showing the performance of MoE in different layers, but they do not explain why the second temporal layer is the optimal choice from a theoretical or neuroscience-informed perspective. Is it because this layer captures high-level, abstract temporal patterns that benefit from specialization?

The baselines chosen for comparison are not representative of the current cutting edge. Many other recent, high-performing models, e.g., other 2024/2025 Transformer variants, Mamba-based models like the one cited as "SUMamba (2026)",  are mentioned in the related work but are not included in the experimental comparisons.

While the paper includes an ablation study, it is not thorough enough to fully validate the design, e.g., why this specific architecture, and why using the staged training scheme, which needs to be further justified.

---

> ### Author Response · Authors · 2025-11-25
> **Response to Reviewer ideL [1/n]**
>
> **The introduction claims that current methods struggle to "jointly exploit local inductive biases and global dependencies." This statement is too general and high-level. It doesn't clearly pinpoint the specific shortcomings of existing models. For example, what exact "local inductive biases" are current CNNs missing? How do existing Transformers fail to capture "global dependencies" in the specific context of EEG signals? A more concrete problem definition is needed.**
>
> We thank the reviewer for this insightful comment. In response, we have substantially revised the relevant portion of the introduction to provide a clearer and more technically grounded explanation of the limitations of existing approaches. Rather than relying on high-level descriptions, the revised text now explicitly identifies the model-specific shortcomings that motivate our proposed method.
>
> First, we clarify the **local inductive bias limitation of CNNs** by noting that their fixed and relatively small receptive fields restrict their ability to model the long-range temporal structure and harmonic relationships that are critical to dense-frequency SSVEP decoding. These details directly address the reviewer’s request for specificity regarding which types of local structure CNNs fail to capture.
>
> Second, we explain that although Transformers are theoretically well-suited for modeling global dependencies, their practical use in SSVEP literature is **predominantly limited to LOSO (leave-one-subject-out) settings**, where global cross-subject patterns are the primary source of generalization. In **individually calibrated** scenarios (our work), this line of work of Transformers often struggles to extract the fine-grained, local frequency-specific patterns required for high-precision decoding, and typically underperforms despite its expressive capacity. This concretely answers the reviewer’s question about how Transformers fail in the EEG context relevant to our problem.
>
> We believe this revision provides the detailed and problem-specific context the reviewer requested. The revised paragraph is reproduced below for convenience:
>
> “To mitigate these issues, convolutional neural networks (CNNs) and, more recently, Transformer-based architectures have been explored for learning richer spatio-temporal representations from EEG, achieving improvements over traditional baselines [1,2]. However, both families of models exhibit concrete gaps when applied to SSVEP decoding. CNNs depend on fixed, small receptive fields, which limits their ability to capture long-range temporal structure and harmonic relationships across wider time-scales features that are essential for differentiating densely spaced SSVEP frequencies. While Transformers are in principle capable of modeling global dependencies, existing works primarily deploy them in leave-one-subject-out (LOSO) evaluations, where broad cross-subject scenario is beneficial. In individually calibrated settings, however,  this line of research often struggles to extract the fine-grained, local frequency-specific patterns required for high-precision decoding, and typically underperforms despite its expressive capacity.”
>
> We hope this resolves the concern by offering a clear, technical justification for the motivation behind our model.
>
> [1] Song et al.  EEG Conformer: Convolutional Transformer for EEG Decoding and Visualization. IEEE Transactions on Neural Systems and Rehabilitation Engineering, 31:710–719, 2022.
>
> [2] Li et al.  Convolutional Correlation Analysis for Enhancing the Performance of SSVEP-Based Brain-Computer Interface. IEEE Transactions on Neural Systems and Rehabilitation Engineering, 28(12):2681–2690, 2020.
>
> Best regards,
>
> The Authors

---

> ### Author Response · Authors · 2025-11-25
> **Response to Reviewer ideL [2/n]**
>
> **Furthermore, the literature review fails to capture the most recent advancements in the field. The most recent related work cited is from 2022, which is a considerable gap in the fast-moving area of deep learning for BCIs. For a paper submitted to ICLR 2026, it is expected to engage with work from 2023 and 2025. The absence of these references raises a critical question: Are the authors solving a problem that has already been addressed by more recent models? Without comparing against these newer methods, it's impossible to know if VIBE is truly state-of-the-art or simply reinventing the wheel.**
>
> We sincerely thank the reviewer for highlighting the importance of including the most recent advancements in our literature review. To address this concern, we have updated our results table to incorporate three additional models from 2023–2025 that represent some of the latest approaches in the field. While we observe that these three models perform similarly to DNN/TRCANet, which was already included as a baseline in our original comparisons, we emphasize that VIBE continues to significantly outperform all these baselines. This updated comparison clearly demonstrates that VIBE achieves state-of-the-art performance and addresses challenges that have not been fully resolved by recent methods. We will include a detailed table in the revised manuscript summarizing these comparisons for clarity.
>
> "Tables summarize the performance of our model compared to baseline methods under different time lengths (0.2 to 1.0 s) on both the Benchmark and beta datasets. Tables 1 and 3 report classification accuracy (\%), while Tables 2 and 4 present ITR. Across all conditions, our model consistently outperforms all baseline methods, demonstrating superior accuracy and efficiency. Additionally, we include comparisons with three recently proposed methods, Dis-ComNet[1],ConsenNet[2], and  SESCNN[3],  which were not included in the result Fig. Our model achieves the highest accuracy and competitive iteration performance across all settings, highlighting its effectiveness and robustness."
>
> **Classification accuracy (\%) of various methods at different time lengths (0.2 - 1.0 s) on the Benchmark dataset. Values are reported as mean $\pm$ standard deviation.**
>
> | Method       | 0.2           | 0.3           | 0.4           | 0.5           | 0.6           | 0.7           | 0.8           | 0.9           | 1.0           |
> |--------------|---------------|---------------|---------------|---------------|---------------|---------------|---------------|---------------|---------------|
> | DNN          | 58.3 ± 3.2    | 72.3 ± 3.0    | 80.8 ± 2.7    | 85.1 ± 2.5    | 88.6 ± 2.2    | 91.6 ± 1.8    | 93.7 ± 1.6    | 95.2 ± 1.3    | 95.8 ± 1.1
> | TRCANet      | 58.4 ± 3.1    | 72.3 ± 3.0    | 80.8 ± 2.7    | 85.2 ± 2.5    | 88.8 ± 2.2    | 91.5 ± 1.8    | 93.7 ± 1.6    | 95.2 ± 1.3    | 95.5 ± 1.1    |
> | Dis-ComNet   | 54.2 ± 2.9    | 68.4 ± 3.6    | 76.0 ± 3.1    | 82.6 ± 3.0    | 86.3 ± 2.7    | 89.1 ± 2.8    | 92.2 ± 2.2    | 93.0 ± 1.9    | 95.6 ± 1.8    |
> | SESCNN       | 60.1 ± 3.2    | 73.5 ± 2.9    | 81.7 ± 3.3    | 85.6 ± 2.9    | 88.7 ± 2.6    | 91.2 ± 2.6    | 93.6 ± 2.7    | 94.9 ± 2.5    | 95.6 ± 2.7    |
> | Ours         | 65.5 ± 3.2    | 77.0 ± 2.9    | 83.8 ± 2.5    | 88.3 ± 2.2    | 91.7 ± 2.1    | 94.2 ± 1.9    | 95.6 ± 1.6    | 96.8 ± 1.1    | 97.4 ± 0.7    |
>
> **ITR (bits/min) of various methods at different time lengths (0.2 - 1.0 s) on the Benchmark dataset. Values are reported as mean $\pm$ standard deviation.**
>
> | Method       | 0.2             | 0.3             | 0.4             | 0.5             | 0.6             | 0.7             | 0.8             | 0.9             | 1.0             |
> |--------------|-----------------|-----------------|-----------------|-----------------|-----------------|-----------------|-----------------|-----------------|-----------------|
> | DNN          | 194.6 ± 14.8    | 235.4 ± 13.5    | 248.8 ± 11.8    | 243.3 ± 10.2    | 236.0 ± 8.5     | 228.1 ± 6.9     | 218.9 ± 5.8     | 208.7 ± 4.4     | 196.5 ± 3.8     |
> | TRCANet      | 196.3 ± 14.8    | 235.5 ± 13.5    | 248.6 ± 11.8    | 243.2 ± 10.2    | 236.2 ± 8.5     | 226.9 ± 6.8     | 217.8 ± 5.8     | 207.6 ± 4.4     | 194.8 ± 3.7     |
> | Dis-ComNet   | 185.2 ± 15.1    | 225.6 ± 13.8    | 236.0 ± 13.5    | 234.3 ± 10.3    | 230.2 ± 9.8     | 220.9 ± 8.3     | 210.3 ± 7.4     | 205.7 ± 6.9     | 198.1 ± 5.4     |
> | SESCNN       | 204.5 ± 14.5    | 237.8 ± 15.9    | 253.0 ± 12.8    | 245.4 ± 11.4    | 236.6 ± 10.7    | 229.3 ± 9.5     | 212.2 ± 7.1     | 204.1 ± 7.4     | 189.9 ± 6.1     |
> | Ours         | 232.8 ± 15.7    | 259.4 ± 14.1    | 263.8 ± 11.7    | 257.7 ± 9.4     | 249.6 ± 8.9     | 238.8 ± 7.2     | 226.1 ± 6.2     | 214.5 ± 5.1     | 202.5 ± 4.2     |
>
>
> More tables continued on next page response

---

> ### Author Response · Authors · 2025-11-25
> **Response to Reviewer ideL [3/n]**
>
> **Furthermore, the literature review fails to capture the most recent advancements in the field. The most recent related work cited is from 2022, which is a considerable gap in the fast-moving area of deep learning for BCIs. For a paper submitted to ICLR 2026, it is expected to engage with work from 2023 and 2025. The absence of these references raises a critical question: Are the authors solving a problem that has already been addressed by more recent models? Without comparing against these newer methods, it's impossible to know if VIBE is truly state-of-the-art or simply reinventing the wheel.**
>
> Accuracy / ITR of various methods at different time lengths (0.2 - 1.0 s) on the BETA dataset. Values are reported as mean $\pm$ standard deviation. ConsenNet was evaluated under a slightly different setting: all but one subject were used as their training set. In their fine-tuning stage, they used the first three calibration blocks from the new subject as training and the remaining blocks as testing. Thus, for each test trial in the BETA dataset, all other trials from that subject, as well as all data from other subjects, were seen during training. They used more data for training compared to our setup, incorporating all other trials from the test subject as well as data from all remaining subjects.
> | Method       | 0.2             | 0.3             | 0.4             | 0.5             | 0.6             | 0.7             | 0.8             | 0.9             | 1.0             |
> |--------------|-----------------|-----------------|-----------------|-----------------|-----------------|-----------------|-----------------|-----------------|-----------------|
> | DNN          | 46.2 ± 2.1      | 59.9 ± 2.1      | 68.0 ± 2.1      | 71.8 ± 2.0      | 75.5 ± 2.0      | 78.7 ± 1.9      | 80.9 ± 1.7      | 82.3 ± 1.7      | 83.7 ± 1.6      |
> | TRCANet      | 46.1 ± 2.1      | 60.0 ± 2.1      | 67.9 ± 2.1      | 71.7 ± 2.0      | 75.6 ± 2.0      | 78.9 ± 1.9      | 80.9 ± 1.7      | 82.2 ± 1.7      | 83.8 ± 1.7      |
> | SESCNN       | 46.9 ± 3.2      | 58.9 ± 3.8      | 68.1 ± 3.5      | 71.2 ± 3.9      | 75.9 ± 4.1      | 79.1 ± 3.4      | 81.1 ± 4.3      | 82.8 ± 3.9      | 84.1 ± 4.7      |
> | ConsenNet    | -               | -               | 67.9 ± 2.1      | -               | 77.7 ± 2.2      | -               | 83.6 ± 1.9      | -               | -               |
> | Ours         | 54.1 ± 2.3      | 64.6 ± 2.0      | 70.8 ± 1.9      | 75.7 ± 1.9      | 79.4 ± 1.8      | 82.2 ± 1.8      | 84.2 ± 1.7      | 85.7 ± 1.6      | 86.3 ± 1.4      |
>
>
> | Method       | 0.2             | 0.3             | 0.4             | 0.5             | 0.6             | 0.7             | 0.8             | 0.9             | 1.0             |
> |--------------|-----------------|-----------------|-----------------|-----------------|-----------------|-----------------|-----------------|-----------------|-----------------|
> | DNN          | 46.2 ± 2.1      | 59.9 ± 2.1      | 68.0 ± 2.1      | 71.8 ± 2.0      | 75.5 ± 2.0      | 78.7 ± 1.9      | 80.9 ± 1.7      | 82.3 ± 1.7      | 83.7 ± 1.6      |
> | TRCANet      | 46.1 ± 2.1      | 60.0 ± 2.1      | 67.9 ± 2.1      | 71.7 ± 2.0      | 75.6 ± 2.0      | 78.9 ± 1.9      | 80.9 ± 1.7      | 82.2 ± 1.7      | 83.8 ± 1.7      |
> | SESCNN       | 46.9 ± 3.2      | 58.9 ± 3.8      | 68.1 ± 3.5      | 71.2 ± 3.9      | 75.9 ± 4.1      | 79.1 ± 3.4      | 81.1 ± 4.3      | 82.8 ± 3.9      | 84.1 ± 4.7      |
> | ConsenNet    | -               | -               | 67.9 ± 2.1      | -               | 77.7 ± 2.2      | -               | 83.6 ± 1.9      | -               | -               |
> | Ours         | 54.1 ± 2.3      | 64.6 ± 2.0      | 70.8 ± 1.9      | 75.7 ± 1.9      | 79.4 ± 1.8      | 82.2 ± 1.8      | 84.2 ± 1.7      | 85.7 ± 1.6      | 86.3 ± 1.4      |
>
> [1] Li et al. Enhancing detection of SSVEPs using discriminant compacted network. Journal of Neural Engineering, 22(1), 016043, 2025.
>
> [2] Zhang  et al. Enhancing SSVEP-Based BCI Performance via Consensus Information Transfer Among Subjects. IEEE Transactions on Neural Networks and Learning Systems, 2024.
>
> [3] Jin  et al. Squeeze and excitation-based multiscale CNN for classification of steady-state visual evoked potentials. IEEE Internet of Things Journal, 2024.
>
> Best regards,
>
> The Authors

---

> ### Author Response · Authors · 2025-11-25
> **Response to Reviewer ideL [4/n]**
>
> **The paper combines several existing components，a Vision Transformer (ViT), a Mixture of Experts (MoE) layer, and data augmentation, but the novelty of this combination is not well-justified. The idea of using a ViT to generate or extend a temporal sequence is interesting. However, the paper does not sufficiently explain why this is a better approach than other sequence generation or feature enhancement techniques. It feels more like an application of an existing architecture ("let's try ViT on EEG") rather than a principled innovation. The ablation study shows it helps, but the underlying "why" remains unclear.**
>
> We thank the reviewer for raising thoughtful concerns regarding the justification of our architectural design and the novelty of combining ViT, MoE, and temporal data extension.
>
> While we agree that ViT, MoE, and augmentation have each appeared independently in prior EEG and sequence-modeling work, our contribution does not lie in merely assembling these components. Rather, the novelty of VIBE emerges from how these modules are reorganized into a staged, interdependent pipeline, where each stage performs a representational transformation required by the next.
>
> Importantly, our use of ViT differs fundamentally from previous EEG applications of transformers. Existing ViT-based EEG models almost universally deploy transformers as **feature extractors**: they encode spatial or spatiotemporal tokens into a compact latent space for classification, similar to **BERT-style token encoders**. Even recent EEG foundation models use transformers solely as signal compressors, not as sequence generators. In contrast, our Stage-1 module employs a ViT as a temporal generator, producing an extended temporal representation from a short EEG input window. This is not simply “trying ViT on EEG”: it enables a capability, where effective temporal-length expansion without increasing acquisition latency, that no prior EEG transformer architecture supports. This addresses a longstanding practical limitation of SSVEP BCIs, where longer windows improve accuracy but slow real-time operation.
>
> Similarly, our MoE plays a distinct role from prior work such as EvoMoE. Earlier MoE-EEG approaches introduce large MoE blocks to learn **global biological structure** or broad population-level variability. In VIBE, MoE is used in a lightweight, localized manner within Conv2D layers to enable frequency-specialized expert routing, a targeted mechanism aligned with the structure of SSVEP signals. Instead of discovering global EEG distributions, our MoE focuses on fine-grained specialization while remaining computationally efficient, a different architectural philosophy from “full-decoder” MoE designs in prior EEG studies.
>
> Together, the ViT-based temporal generator and the lightweight frequency-specialized MoE form the backbone of VIBE’s four-stage pipeline which is designed sequentially rather than as a set of interchangeable modules. This organization and functional differentiation represent a principled architectural contribution rather than a simple recombination of existing techniques.
>
> To evaluate our choice of ViT for temporal sequence generation, we conducted a controlled comparison with another transformer, the Convolutional Vision Transformer (**CvT**), while keeping all other VIBE components identical. The results in Table show that ViT consistently outperforms CvT across multiple benchmarks. This demonstrates that the improved performance of VIBE is not just due to using a transformer, but specifically arises from the ViT-based generator, which effectively extends short EEG segments and preserves critical frequency-specific information. These findings provide a clear justification for our design choice and show that ViT offers a practical advantage over other sequence generation approaches. A section named Comparison of Transformer-Based Generators for Temporal Sequence Extension is added in the revision.
>
> "To rigorously evaluate the design choice of using a ViT for temporal sequence generation, we conducted a comparative study against an alternative transformer-based architecture, the Convolutional Vision Transformer (CvT). Both models were tasked with generating extended temporal sequences from short EEG segments of 0.2 s, while keeping all modules in multistages identical. This controlled setup isolates the effect of the generator architecture on the overall decoding performance. Quantitative results, summarized in Table, demonstrate that ViT consistently achieves higher mean accuracy in two datasets. These findings substantiate the selection of ViT as the temporal generator in VIBE, confirming its advantage for extending short EEG segments while preserving fine-grained, frequency-specific information critical for high-precision SSVEP decoding. "
>
> Table continued on next page response

---

> ### Author Response · Authors · 2025-11-25
> **Response to Reviewer ideL [5/n]**
>
> **The paper combines several existing components，a Vision Transformer (ViT), a Mixture of Experts (MoE) layer, and data augmentation, but the novelty of this combination is not well-justified. The idea of using a ViT to generate or extend a temporal sequence is interesting. However, the paper does not sufficiently explain why this is a better approach than other sequence generation or feature enhancement techniques. It feels more like an application of an existing architecture ("let's try ViT on EEG") rather than a principled innovation. The ablation study shows it helps, but the underlying "why" remains unclear.**
>
>
> Comparison of ViT and CvT as temporal sequence generators. All other components in the VIBE architecture remain fixed. The results show that ViT consistently outperforms CvT in mean accuracy, supporting its selection as the temporal generator
>
> | Model | Benchmark 0.2s | Benchmark 0.4s | BETA 0.2s | BETA 0.4s |
> |-------|----------------|----------------|------------|------------|
> | ViT   | 65.5           | 83.8           | 54.1       | 70.8       |
> | CvT   | 62.1           | 82.5           | 51.0       | 69.1       |
>
> Best regards,
>
> The Authors

---

> ### Author Response · Authors · 2025-11-25
> **Response to Reviewer ideL [6/n]**
>
> **The decision to replace the last temporal convolution layer with an MoE layer is a major architectural choice, but it is made without strong justification. The authors provide an ablation table in appendix Table 2, showing the performance of MoE in different layers, but they do not explain why the second temporal layer is the optimal choice from a theoretical or neuroscience-informed perspective. Is it because this layer captures high-level, abstract temporal patterns that benefit from specialization?**
>
> To address the reviewer’s concern regarding the placement of the MoE in the second temporal convolution layer, we conducted a focused interpretability analysis on the fine-tuned decoder. We applied Grad-CAM [1] to the final MoE temporal layer to visualize expert-specific temporal saliency. The resulting temporal maps were averaged across channels, zero-padded to 5 s for uniform spectral resolution, and analyzed in both time and frequency domains via FFT. In addition, we tracked expert-selection counts per target frequency to quantify how responsibility is distributed across experts.
>
> The analysis shows that the MoE does not collapse to a single expert; instead, different target frequencies consistently activate distinct experts, demonstrating that the layer specializes in diverse temporal–spectral structures rather than redundant patterns. The Grad-CAM frequency analysis further reveals strong peaks at the stimulus frequency and its harmonics, confirming that the layer captures canonical SSVEP dynamics rather than dataset-specific artifacts. These results provide direct evidence that the second temporal layer with MoE effectively decomposes EEG temporal dynamics into meaningful components, justifying its placement both theoretically and empirically.
>
> A section name Visualization in MoE Temporal Layer is added in Appendix.
>
> "We perform a qualitative analysis of the MoE temporal layer using the fine-tuned decoder for a randomly selected subject. Grad-CAM [1] is applied to the final MoE temporal convolution layer to obtain activation importance over time. In Figure, the resulting temporal map is averaged across channels to produce a single 1D sequence. As in the previous analyses, this sequence is zero-padded to 5s to ensure sufficient spectral resolution before applying the FFT. In addition to the frequency-domain visualization, we record the MoE expert selection count Figure for each target frequency to examine how the mixture-of-experts distributes attention across different spectral components.
>
> The MoE expert selection patterns reveal that the experts do not collapse into a single dominant expert; instead, different target frequencies elicit distinct expert activation distributions. This diversity indicates that individual experts specialize in different temporal–spectral patterns rather than redundantly modeling the same structure. Complementing this, the Grad-CAM analysis shows clear spectral peaks at the target frequency and its harmonic after FFT, demonstrating that the MoE temporal layer effectively pools and amplifies frequency-specific structure. Together, these results confirm that the MoE architecture meaningfully decomposes the temporal dynamics and contributes specialized processing across targets."
>
> [1] Selvaraju et al. Grad-cam: Visual explanations from deep networks via gradient-based localization. ICCV, 2017.
>
> Best regards,
>
> The Authors

---

> ### Author Response · Authors · 2025-11-25
> **Response to Reviewer ideL [7/n]**
>
> **While the paper includes an ablation study, it is not thorough enough to fully validate the design, e.g., why this specific architecture, and why using the staged training scheme, which needs to be further justified.**
>
> We thank the reviewer for highlighting the need for a clearer justification of VIBE’s staged design. In the revised manuscript, we explicitly clarify both the theoretical motivation and the empirical necessity of each stage.
>
> From a theoretical perspective, we observe that with each additional stage, the activations in the final temporal layer exhibit increasingly stronger signals at the target frequency and its harmonics. This indicates that each stage progressively enhances the frequency-specific representation crucial for SSVEP decoding, providing a principled basis for the staged design.
>
> To rigorously evaluate the contribution of each stage, we introduced a new analysis in the revision: “Evaluating Stage Contributions via MoE Temporal Layer Ablation.” This study examines how each stage affects the learned temporal–spectral structure by computing FFT-based SNR directly on the MoE temporal layer activations. We performed a controlled comparison across four conditions: Stage 3 vs. Stage 4 decoders and ViT-augmented vs. original-only inputs, while keeping all other architectural and data settings constant.
> The results (Figure and Table ) confirm that the components previously questioned produce statistically significant improvements in the frequency-domain structure. Target-frequency SNR consistently increases across conditions from BASE-original → BASE-full → FT-original → FT-full, with all paired comparisons reaching significance. These findings demonstrate that each stage makes a measurable, nonredundant contribution to decoding performance, confirming that the staged design is both theoretically motivated and empirically necessary for high-precision SSVEP decoding.
>
>
> **Paired t-test results comparing SNR differences from the final temporal layer activations. FT refers to Stage 4 and BASE refers to Stage 3. Full uses the complete input sequence (original data + Stage 2 ViT-generated data), while Original keeps only the original data and zero-pads the rest.**
>
> | Comparison                  | t-stat | p-value     | n  |
> | --------------------------- | ------ | ----------- | -- |
> | FT (full − original)        | 3.3941 | 1.59×10^-3  | 40 |
> | BASE (full − original)      | 3.0741 | 3.84×10^-3  | 40 |
> | FT full − BASE full         | 9.2720 | 2.08×10^-11 | 40 |
> | FT original − BASE original | 7.6147 | 3.12×10^-9  | 40 |
>
>
>
> In the Appendix section Evaluating Stage Contributions via MoE Temporal Layer Ablation,  we wrote:
>
> "In this analysis, we investigate how different stages of VIBE influence the model’s frequency domain behavior under a controlled experimental setting. We use slightly modified parameters relative to the main experiments: each input consists of a 1s EEG segment, and the Stage~1 ViT generator produces an additional 0.2s extension. The decoder is implemented with eight temporal experts in the MoE layer. Both the Stage 3 (non-finetuned) and Stage 4 (finetuned) decoders are trained using the ViT augmented data, and all visualizations are performed using the same training set to remove cross-split variability. For each subject and trial, we extract the activation from the final temporal Conv2D layer and average across channels as well as subjects to obtain a single temporal sequence per target. This sequence is zero-padded to 5s to achieve sufficient spectral resolution for targets ranging from 8 Hz to 15.8 Hz, after which we apply an FFT. We compute the signal-to-noise ratio (SNR) by taking the peak amplitude at the target frequency and its second harmonic, normalizing each by the mean amplitude of neighboring frequency bins, and averaging the two values. We then perform paired t-tests to assess statistical significance. The analysis compares four conditions: (1) Stage 3 versus Stage 4 decoders, and (2) ViT augmented input (original+generated) versus original only input with zero-padding. The corresponding FFT visualizations and t-test statistics are summarized in the accompanying Figure  and Table
> The FFT visualizations and t-test results consistently show that both the inclusion of ViT-generated data and decoder fine-tuning improve spectral responses. Specifically, the amplitude at the target frequency and its second harmonic is highest for FT with full input, followed by FT with original input, then BASE full, and finally BASE original. All comparisons are statistically significant, highlighting that ViT augmentation and Stage 4 fine-tuning are important for enhancing frequency-specific features in the model."
>
> Best regards,
>
> The Authors

---

> ### Author Response · Authors · 2025-11-28
> **Response to Reviewer ideL [8/n]**
>
> Dear Reviewer ideL,
>
> Thank you for your thoughtful and constructive comments. We have carefully addressed all points raised, and the revised submission now reflects all corresponding updates and clarifications. Please feel free to provide any additional feedback or suggestions. We appreciate your time and effort in helping us improve our work.
>
> Best regards,
>
> The Authors

---

> ### Author Response · Authors · 2025-12-01
> **Response to Reviewer ideL [9/n]**
>
> **The baselines chosen for comparison are not representative of the current cutting edge. Many other recent, high-performing models, e.g., other 2024/2025 Transformer variants, Mamba-based models like the one cited as "SUMamba (2026)", are mentioned in the related work but are not included in the experimental comparisons.**
>
> We thank the reviewer for highlighting the importance of comparing against the most recent high-performing models. While we acknowledge that models such as the 2024/2025 Transformer variants and Mamba-based architectures (e.g., SUMamba (2026)) represent the cutting edge, we note that SUMamba employs a fundamentally different experimental setting,specifically, a leave-one-subject-out (LOSO) evaluation, whereas our study focuses on individually calibrated decoding. Direct comparison under these differing settings would be misleading and is therefore deferred as future work.
>
>
> Best regards,
>
> The Authors

---

### Official Review · Reviewer_S2CM · 2025-10-28

**Soundness:** 3
**Presentation:** 2
**Contribution:** 3
**Rating:** 6
**Confidence:** 4

**Summary:**

The paper proposes a framework to address the SSVEP EEG classification task. It features a cleverly-designed ViT encoder-decoder structure, alongside a MoE, and several pre-training stages, that all-together result in good classification and efficiency performances. The paper is well-written but seems to lack some ablation studies (see below), some choices of hyperparameters are not specified, and the structure of the Method section is a bit hectic to go through sequentially. There are also some interesting introductions of data augmentation techniques.

**Strengths:**

- performances above current SotA on both datasets.

- the introduction of an auxiliary load-balancing loss, which prevents the model from sticking to the same subset of experts in the MoE throughout training.

- a series of t-SNE plots in the appendix to visualize the knowledge of the trained proposed model.

- computational stats are provided in Tab. 7, such as training and test time. Indeed, the proposed approach seems very efficient in both phases.

- code is provided to reproduce results.

**Weaknesses:**

- there is no "results table", showing the numbers obtained by the proposed method compared to the rest of the related methods. What comes close are Fig. 2 and 3, but the individual numbers are not shown for each point in the plots.

- the ablation study lacks some info, like the results of the baseline with/without all the data augmentation techniques to which to compare in Tab. 3. Moreover, no results about combination of techniques are provided (e.g. model using only stitching and channel shuffle, or model using only channel shuffle).

- in the paragraph that starts at L656, it is not clear why such values for learning rate or batch size are used.

- Fig. 1 is not so auto-explicative. This reviewer believe that there is too much text in it, without a clear indication about where to start looking at. Moreover, the caption only talks about points a) and e), but not b-d). I suggest redesigning the figure and its caption entirely.

**Questions:**

- it seems interesting the authors are merging different chunks of EEGs in the cross-subject temporal stitching. But what is the rationale behind this choice? Creating new samples by concatenating two chunks pertaining to (eventually) different subjects or trials without accounting for inter-subject/trial variabilities may lead to meaningless new samples.

- This reviewer gets the idea behind channel chunk shuffle, but intuitively it may completely alter the meaning of an EEG: think about how randomly swapping the RGB channels of a natural image completely alters its colors. Again, what is the rationale?

- why did the authors choose values of 6 and 4 k-folds for the two datasets? If it is due to replicating the setting of another method, there should be a citation on L305.

---

> ### Author Response · Authors · 2025-11-26
> **Response to Reviewer S2CM [1/n]**
>
> **Weakness 1. there is no "results table", showing the numbers obtained by the proposed method compared to the rest of the related methods. What comes close are Fig. 2 and 3, but the individual numbers are not shown for each point in the plots.**
>
>
> We thank the reviewer for the feedback. In the revision, we have added explicit results tables that report the numerical values of classification accuracy and ITR for our proposed method alongside all baseline methods. These tables complement Figs. 2 and 3, providing the precise numbers for each point in the plots and enabling direct quantitative comparison across methods. A new section name Results is added in appendix, section A.10
>
> Classification accuracy (\%) of various methods at different time lengths (0.2 - 1.0 s) on the Benchmark dataset. Values are reported as mean $\pm$ standard deviation.
>
> | Method        | 0.2           | 0.3           | 0.4           | 0.5           | 0.6           | 0.7           | 0.8           | 0.9           | 1.0           |
> |---------------|---------------|---------------|---------------|---------------|---------------|---------------|---------------|---------------|---------------|
> | DNN           | 58.3 ± 3.2    | 72.3 ± 3.0    | 80.8 ± 2.7    | 85.1 ± 2.5    | 88.6 ± 2.2    | 91.6 ± 1.8    | 93.7 ± 1.6    | 95.2 ± 1.3    | 95.8 ± 1.1    |
> | TDCA          | 54.1 ± 3.4    | 70.5 ± 3.4    | 79.1 ± 3.1    | 84.6 ± 2.7    | 89.1 ± 2.3    | 92.2 ± 2.0    | 94.3 ± 1.6    | 95.7 ± 1.2    | 96.6 ± 1.0    |
> | TRCA          | 49.0 ± 3.6    | 64.0 ± 3.6    | 72.5 ± 3.5    | 79.3 ± 3.2    | 84.1 ± 3.0    | 87.6 ± 2.8    | 90.2 ± 2.5    | 92.5 ± 2.1    | 93.6 ± 1.9    |
> | eCCA          | 28.3 ± 2.7    | 49.4 ± 3.7    | 62.7 ± 3.9    | 72.7 ± 3.6    | 79.9 ± 3.3    | 85.4 ± 3.0    | 89.4 ± 2.6    | 92.1 ± 2.1    | 93.5 ± 1.8    |
> | msTRCA        | 53.3 ± 3.4    | 66.7 ± 3.6    | 74.3 ± 3.4    | 79.8 ± 3.2    | 84.4 ± 3.0    | 88.4 ± 2.7    | 90.7 ± 2.4    | 92.7 ± 2.0    | 93.9 ± 1.8    |
> | SSVEPformer   | 46.1 ± 3.1    | 62.0 ± 3.5    | 69.7 ± 3.3    | 76.3 ± 3.1    | 81.5 ± 3.0    | 86.0 ± 2.7    | 89.7 ± 2.4    | 91.9 ± 2.0    | 93.0 ± 1.8    |
> | TRCANet       | 58.4 ± 3.1    | 72.3 ± 3.0    | 80.8 ± 2.7    | 85.2 ± 2.5    | 88.8 ± 2.2    | 91.5 ± 1.8    | 93.7 ± 1.6    | 95.2 ± 1.3    | 95.5 ± 1.1    |
> | Dis-ComNet    | 54.2 ± 2.9    | 68.4 ± 3.6    | 76.0 ± 3.1    | 82.6 ± 3.0    | 86.3 ± 2.7    | 89.1 ± 2.8    | 92.2 ± 2.2    | 93.0 ± 1.9    | 95.6 ± 1.8    |
> | SESCNN        | 60.1 ± 3.2    | 73.5 ± 2.9    | 81.7 ± 3.3    | 85.6 ± 2.9    | 88.7 ± 2.6    | 91.2 ± 2.6    | 93.6 ± 2.7    | 94.9 ± 2.5    | 95.6 ± 2.7    |
> | Ours          | 65.5 ± 3.2    | 77.0 ± 2.9    | 83.8 ± 2.5    | 88.3 ± 2.2    | 91.7 ± 2.1    | 94.2 ± 1.9    | 95.6 ± 1.6    | 96.8 ± 1.1    | 97.4 ± 0.7    |
>
> ITR  (bits/min) of various methods at different time lengths (0.2 - 1.0 s) on the Benchmark dataset. Values are reported as mean $\pm$ standard deviation.
>
> | Method        | 0.2           | 0.3           | 0.4           | 0.5           | 0.6           | 0.7           | 0.8           | 0.9           | 1.0           |
> |---------------|---------------|---------------|---------------|---------------|---------------|---------------|---------------|---------------|---------------|
> | DNN           | 194.6 ± 14.8  | 235.4 ± 13.5  | 248.8 ± 11.8  | 243.3 ± 10.2  | 236.0 ± 8.5   | 228.1 ± 6.9   | 218.9 ± 5.8   | 208.7 ± 4.4   | 196.5 ± 3.8   |
> | TDCA          | 173.9 ± 15.5  | 227.5 ± 15.3  | 240.6 ± 13.0  | 240.5 ± 11.0  | 237.3 ± 8.8   | 230.3 ± 7.4   | 220.8 ± 5.8   | 210.0 ± 4.4   | 199.1 ± 3.4   |
> | msTRCA        | 151.7 ± 16.1  | 197.3 ± 15.6  | 212.0 ± 14.2  | 218.5 ± 12.3  | 217.7 ± 10.7  | 213.4 ± 9.5   | 206.3 ± 8.0   | 199.2 ± 6.6   | 189.5 ± 5.7   |
> | eCCA          | 65.2 ± 10.0   | 134.8 ± 14.9  | 171.4 ± 15.0  | 192.2 ± 13.4  | 201.6 ± 11.7  | 205.2 ± 10.1  | 203.3 ± 8.3   | 197.8 ± 6.7   | 188.9 ± 5.6   |
> | msTRCA        | 170.5 ± 16.0  | 210.1 ± 15.8  | 220.1 ± 14.0  | 220.6 ± 12.5  | 219.5 ± 10.9  | 216.7 ± 9.4   | 208.2 ± 7.9   | 199.9 ± 6.5   | 190.6 ± 5.6   |
> | SSVEPformer   | 137.4 ± 13.9  | 188.8 ± 15.3  | 199.3 ± 13.5  | 206.2 ± 11.9  | 208.0 ± 10.7  | 207.0 ± 9.2   | 204.4 ± 7.8   | 197.5 ± 6.4   | 187.9 ± 5.6   |
> | TRCANet       | 196.3 ± 14.8  | 235.5 ± 13.5  | 248.6 ± 11.8  | 243.2 ± 10.2  | 236.2 ± 8.5   | 226.9 ± 6.8   | 217.8 ± 5.8   | 207.6 ± 4.4   | 194.8 ± 3.7   |
> | Dis-ComNet    | 185.2 ± 15.1  | 225.6 ± 13.8  | 236.0 ± 13.5  | 234.3 ± 10.3  | 230.2 ± 9.8   | 220.9 ± 8.3   | 210.3 ± 7.4   | 205.7 ± 6.9   | 198.1 ± 5.4   |
> | SESCNN        | 204.5 ± 14.5  | 237.8 ± 15.9  | 253.0 ± 12.8  | 245.4 ± 11.4  | 236.6 ± 10.7  | 229.3 ± 9.5   | 212.2 ± 7.1   | 204.1 ± 7.4   | 189.9 ± 6.1   |
> | Ours          | 232.8 ± 15.7  | 259.4 ± 14.1  | 263.8 ± 11.7  | 257.7 ± 9.4   | 249.6 ± 8.9   | 238.8 ± 7.2   | 226.1 ± 6.2   | 214.5 ± 5.1   | 202.5 ± 4.2   |
>
> More tables continued on next page response

---

> ### Author Response · Authors · 2025-11-26
> **Response to Reviewer S2CM [2/n]**
>
> **Weakness 1. there is no "results table", showing the numbers obtained by the proposed method compared to the rest of the related methods. What comes close are Fig. 2 and 3, but the individual numbers are not shown for each point in the plots.**
>
> Classification accuracy (%) of various methods at different time lengths (0.2 - 1.0 s) on the BETA dataset. Values are reported as mean $\pm$ standard deviation.
> | Method        | 0.2           | 0.3           | 0.4           | 0.5           | 0.6           | 0.7           | 0.8           | 0.9           | 1.0           |
> |---------------|---------------|---------------|---------------|---------------|---------------|---------------|---------------|---------------|---------------|
> | DNN           | 46.2 ± 2.1    | 59.9 ± 2.1    | 68.0 ± 2.1    | 71.8 ± 2.0    | 75.5 ± 2.0    | 78.7 ± 1.9    | 80.9 ± 1.7    | 82.3 ± 1.7    | 83.7 ± 1.6    |
> | TDCA          | 41.4 ± 2.4    | 55.2 ± 2.8    | 63.3 ± 2.7    | 68.0 ± 2.6    | 72.4 ± 2.4    | 76.2 ± 2.3    | 78.8 ± 2.1    | 80.9 ± 2.0    | 82.9 ± 1.9    |
> | msTRCA        | 36.1 ± 2.2    | 47.7 ± 2.7    | 54.9 ± 2.9    | 59.1 ± 2.9    | 63.4 ± 2.8    | 66.7 ± 2.9    | 69.4 ± 2.8    | 72.2 ± 2.8    | 74.2 ± 2.7    |
> | eCCA          | 27.8 ± 1.8    | 43.2 ± 2.5    | 53.0 ± 2.7    | 60.2 ± 2.7    | 66.2 ± 2.6    | 71.0 ± 2.6    | 74.2 ± 2.5    | 77.3 ± 2.4    | 79.8 ± 2.2    |
> | msTRCA        | 40.0 ± 2.3    | 50.7 ± 2.7    | 57.8 ± 2.7    | 61.5 ± 2.7    | 66.0 ± 2.7    | 69.2 ± 2.7    | 72.0 ± 2.6    | 74.7 ± 2.5    | 76.5 ± 2.5    |
> | SSVEPformer   | 37.8 ± 1.9    | 50.8 ± 2.3    | 58.9 ± 2.4    | 64.7 ± 2.4    | 68.9 ± 2.4    | 73.3 ± 2.3    | 76.5 ± 2.3    | 79.4 ± 2.2    | 81.5 ± 2.1    |
> | TRCANet       | 46.1 ± 2.1    | 60.0 ± 2.1    | 67.9 ± 2.1    | 71.7 ± 2.0    | 75.6 ± 2.0    | 78.9 ± 1.9    | 80.9 ± 1.7    | 82.2 ± 1.7    | 83.8 ± 1.7    |
> | SESCNN        | 46.9 ± 3.2    | 58.9 ± 3.8    | 68.1 ± 3.5    | 71.2 ± 3.9    | 75.9 ± 4.1    | 79.1 ± 3.4    | 81.1 ± 4.3    | 82.8 ± 3.9    | 84.1 ± 4.7    |
> | ConsenNet     | -             | -             | 67.9 ± 2.1    | -             | 77.7 ± 2.2    | -             | 83.6 ± 1.9    | -             | -             |
> | Ours          | 54.1 ± 2.3    | 64.6 ± 2.0    | 70.8 ± 1.9    | 75.7 ± 1.9    | 79.4 ± 1.8    | 82.2 ± 1.8    | 84.2 ± 1.7    | 85.7 ± 1.6    | 86.3 ± 1.4    |
>
> ITR (bits/min) of various methods at different time lengths (0.2 - 1.0 s) on the BETA dataset. Values are reported as mean $\pm$ standard deviation.
> | Method        | 0.2           | 0.3           | 0.4           | 0.5           | 0.6           | 0.7           | 0.8           | 0.9           | 1.0           |
> |---------------|---------------|---------------|---------------|---------------|---------------|---------------|---------------|---------------|---------------|
> | DNN           | 137.2 ± 9.1   | 176.4 ± 9.0   | 190.1 ± 8.5   | 185.6 ± 7.6   | 183.0 ± 7.0   | 178.9 ± 6.3   | 172.0 ± 5.5   | 164.2 ± 5.0   | 157.4 ± 4.6   |
> | TDCA          | 118.3 ± 9.7   | 159.9 ± 11.1  | 173.0 ± 10.1  | 172.8 ± 9.1   | 172.4 ± 8.1   | 170.6 ± 7.3   | 165.5 ± 6.5   | 160.2 ± 5.9   | 154.8 ± 5.2   |
> | msTRCA        | 95.8 ± 8.5    | 129.4 ± 10.2  | 141.8 ± 10.1  | 142.1 ± 9.3   | 143.2 ± 8.7   | 141.8 ± 8.3   | 138.8 ± 7.7   | 136.9 ± 7.2   | 132.8 ± 6.6   |
> | eCCA          | 63.1 ± 6.4    | 110.2 ± 9.1   | 133.6 ± 9.5   | 144.5 ± 9.0   | 151.2 ± 8.3   | 154.3 ± 7.8   | 151.9 ± 7.0   | 150.0 ± 6.5   | 146.8 ± 5.8   |
> | msTRCA        | 111.7 ± 9.0   | 140.5 ± 10.2  | 151.6 ± 9.7   | 149.7 ± 8.9   | 151.2 ± 8.5   | 148.8 ± 7.9   | 145.8 ± 7.3   | 143.0 ± 6.8   | 138.1 ± 6.2   |
> | SSVEPformer   | 101.1 ± 7.4   | 139.7 ± 9.2   | 154.9 ± 8.9   | 160.5 ± 8.4   | 160.6 ± 7.9   | 161.8 ± 7.3   | 159.6 ± 6.6   | 157.3 ± 6.1   | 153.0 ± 5.6   |
> | TRC           | 137.6 ± 9.0   | 176.7 ± 9.0   | 189.9 ± 8.5   | 185.4 ± 7.6   | 183.2 ± 7.0   | 179.9 ± 6.4   | 171.9 ± 5.5   | 163.6 ± 5.0   | 157.7 ± 4.6   |
> | SESCNN        | 141.2 ± 9.5   | 178.5 ± 10.3  | 189.3 ± 9.2   | 186.4 ± 8.4   | 182.8 ± 7.2   | 179.1 ± 6.5   | 171.1 ± 5.9   | 165.8 ± 4.3   | 159.1 ± 4.8   |
> | ConsenNet     | -             | -             | 188.7 ± 9.0   | -             | 191.4 ± 7.7   | -             | 181.8 ± 6.2   | -             | -             |
> | Ours          | 173.4 ± 10.4  | 199.1 ± 10.1  | 202.7 ± 8.9   | 201.9 ± 7.3   | 197.4 ± 7.2   | 191.2 ± 6.8   | 182.9 ± 5.9   | 174.7 ± 4.5   | 164.0 ± 4.1   |
>
> Best regards,
>
> The Authors

---

> ### Author Response · Authors · 2025-11-26
> **Response to Reviewer S2CM [3/n]**
>
> **Weakness 2. the ablation study lacks some info, like the results of the baseline with/without all the data augmentation techniques to which to compare in Tab. 3. Moreover, no results about combination of techniques are provided (e.g. model using only stitching and channel shuffle, or model using only channel shuffle).**
>
> We thank the reviewer for the suggestion. In the revision, we have added additional Table 11  showing the progressive effects of combining augmentation components. Specifically, we now report results from using only decorrelation, decorrelation+stitching, decorrelation+stitching+crop, and decorrelation+stitching+crop+shuffle. These additions provide a clearer view of how each augmentation step contributes to performance and allow for a direct comparison of individual and combined techniques.
>
> **Table: Data Augmentation Ablation Study (Benchmark, 0.2s)**
> Effect of progressively adding augmentation components.
>
> | Augmentation Configuration | Decorrelation | Decorrelation + Stitching | Decorrelation + Stitching + Crop | Decorrelation + Stitching + Crop + Shuffle |
> |---------------------------|---------------|-----------------------------|----------------------------------|---------------------------------------------|
> | **Accuracy (%)**          | 64.1          | 64.7                        | 64.9                             | 65.5                                        |
>
> **Weakness 3: in the paragraph that starts at L656, it is not clear why such values for learning rate or batch size are used.**
>
> We thank the reviewer for pointing out the lack of clarity regarding our training hyperparameters. Specifically, we use a consistent batch size of 32 across all training stages. This setting is chosen to avoid excessive generalization during subject-specific fine-tuning on the Benchmark dataset, where each subject has only 200 calibration samples. Larger batch sizes (e.g., 64 or 128) would substantially reduce gradient diversity and harm adaptation performance under this limited-data regime.
> Regarding the learning rate, we intentionally use different learning rates for the ViT generator and the MoE decoder, as these modules exhibit different optimization dynamics and levels of parameter sensitivity. The ViT component requires a smaller learning rate due to its pretrained initialization and higher parameter capacity, while the decoder benefits from a slightly higher learning rate to adapt effectively during both pretraining and fine-tuning.
>
> **Weakness 4. Fig. 1 is not so auto-explicative. This reviewer believe that there is too much text in it, without a clear indication about where to start looking at. Moreover, the caption only talks about points a) and e), but not b-d). I suggest redesigning the figure and its caption entirely.**
>
> We thank the reviewer for the suggestion. We have fully revised the paragraph. "Framework of the proposed VIBE method. Panel (b) presents the overall pipeline, from short SSVEP EEG segments to final classification. Panel (c) depicts the data preparation strategy used during training and testing. Temporal sequence generation with the ViT is detailed in (a), identical for both pretrain and fine-tune stages. The MoE decoder is shown in (e), also identical for both pretrain and fine-tune stages. Panel (d) illustrates the final convolutional temporal layer integrated with the MoE, highlighting how expert activations contribute to the classification output."
>
> **Question 3. why did the authors choose values of 6 and 4 k-folds for the two datasets? If it is due to replicating the setting of another method, there should be a citation on L305.**
>
> We follow the standard “leave-one-block-out” evaluation strategy for both datasets, which naturally results in 6 and 4 folds for the Benchmark and BETA datasets, respectively. This choice aligns with the proposed validation methods on these datasets[1][2], ensuring comparability with existing results.
>
> [1] Wang et al. A benchmark dataset for SSVEP-based brain–computer interfaces. IEEE Transactions on Neural Systems and Rehabilitation Engineering, 25(10):1746–1752, 2016.
>
> [2] Liu et al. BETA: A large benchmark database toward SSVEP-BCI application. Frontiers in Neuroscience, 14:627, 2020.
>
> Best regards,
>
> The Authors

---

> ### Author Response · Authors · 2025-11-26
> **Response to Reviewer S2CM [4/n]**
>
> **Question 1. it seems interesting the authors are merging different chunks of EEGs in the cross-subject temporal stitching. But what is the rationale behind this choice? Creating new samples by concatenating two chunks pertaining to (eventually) different subjects or trials without accounting for inter-subject/trial variabilities may lead to meaningless new samples.**
>
> We thank the reviewer for this insightful comment. The rationale for Cross-Subject Temporal Stitching is that SSVEPs signal exhibits strong frequency-specific structure that is largely preserved across subjects. By concatenating temporal chunks from different trials or subjects, we increase the diversity of the training data, allowing the model to learn invariant spectral patterns rather than memorizing trial- or subject-specific idiosyncrasies. To ensure that the stitched samples remain meaningful, we performed a spectral validation: after Stage-2 ViT generation and Stitch recomposition, we extracted a single subband, averaged across channels and trials, zero-padded to 5 s, and applied an FFT. The resulting spectra consistently show clear peaks at the target frequency and its harmonics in Figure, demonstrating that the essential SSVEP signatures are preserved. These results confirm that the stitched samples provide neurophysiologically valid training examples while improving the model’s generalization across subjects and trials.
>
> A new section A. 9 in revison named Spectral Validation of Stitch-Augmented Data.
>
> "To assess whether the Cross-Subject Temporal Stitching augmentation preserves useful SSVEP structure, we visualize the temporal and spectral profiles of the augmented samples.We use 1 s temporal length original data as input from Benchmark.  After Stage 2 ViT generation (add 0.2s generate data) and Stitch recomposition, we select a single subband, average across channels and trials dimensions, zero-pad the resulting sequence to 5s, and apply an FFT. The corresponding temporal traces and spectra consistently exhibit clear peaks at the target frequency and its harmonic, indicating that the stitched signals retain the essential SSVEP signatures. Although Stitch mixes temporal chunks across channels and subjects, this variability diversifies the training distribution for the decoder while maintaining the frequency-specific structure required for accurate decoding. The accompanying plot (Figure) contains two subpanels: the left panel shows the stitched time-series signal, and the right panel shows the FFT magnitude spectrum. In the spectral panel, clear peaks appear at the target frequency and its harmonic (2$\times$freq), confirming that the stitched samples preserve the characteristic SSVEP structure."
>
> **Question 2. This reviewer gets the idea behind channel chunk shuffle, but intuitively it may completely alter the meaning of an EEG: think about how randomly swapping the RGB channels of a natural image completely alters its colors. Again, what is the rationale?**
>
> We thank the reviewer for the comment. Each EEG channel exhibits spectral characteristics but does not have a strict sequential ordering (although some channels are dominant) like RGB channels in an image. To improve model generalization, we perform shuffling within defined temporal-spatial chunks, exposing the network to different channel arrangements during training. This procedure preserves the underlying physiological information, as the network focuses on learning invariant spectral patterns rather than depending on fixed channel positions. A new section A.10 named Spectral Validation of Channel Chunk Shuffle Augmented Data is added regarding this issue.
> ‘To verify that the Channel Chunk Shuffle augmentation preserves essential SSVEP structure, we visualized the spectral profiles of augmented samples. We use 1 s temporal length original data as input from Benchmark. Following Stage 2 ViT generation (add 0.5s data) and shuffle recomposition, we select a single subband and channel, average across trials, zero-pad the resulting sequence to 5 s, and compute the FFT. By shuffling within defined temporal chunks, we expose the network to different channel arrangements, enhancing generalization while preserving physiologically meaningful spectral features. The accompanying plot (Figure)shows that prominent peaks at the target frequency and its harmonic, indicating that the shuffled signals maintain the frequency-specific structure required for accurate decoding.
>
> Best regards,
>
> The Authors

---

> ### Author Response · Authors · 2025-11-28
> **Response to Reviewer S2CM [5/n]**
>
> Dear Reviewer S2CM,
>
> Thank you for your thoughtful and constructive comments. We have carefully addressed all points raised, and the revised submission now reflects all corresponding updates and clarifications. Please feel free to provide any additional feedback or suggestions. We appreciate your time and effort in helping us improve our work.
>
> Best regards,
>
> The Authors

---

### Official Review · Reviewer_3S4g · 2025-10-29

**Soundness:** 2
**Presentation:** 1
**Contribution:** 2
**Rating:** 2
**Confidence:** 5

**Summary:**

This paper introduces VIBE (Vision Transformer Based Experts Network), a multistage deep learning framework designed for steady-state visual evoked potential (SSVEP) classification in brain-computer interfaces. The work addresses the performance bottleneck of existing deep learning methods for SSVEP-BCI decoding, which struggle to fully extract complex neural signal features required for robust performance under short time windows. VIBE employs a dual-component architecture consisting of a Vision Transformer (ViT) module for temporal generation and a Mixture of Experts (MoE) decoder for frequency recognition. The ViT module expands short input EEG sequences (multi-band data represented as tensors with dimensions corresponding to sub-bands, channels, and time samples) into richer temporal representations by partitioning signals into non-overlapping patches, processing them through a transformer-based decoder, and concatenating the generated temporal segments with original inputs. The MoE decoder integrates subband, channel, and temporal dependencies through expert specialization, where a gating network routes inputs to top-k experts that process distinct temporal or spectral patterns, enhanced by an auxiliary load-balancing loss using KL divergence to encourage balanced expert utilization and prevent overfitting. The framework incorporates data augmentation strategies specifically designed for EEG, including cross-subject temporal stitching (creating synthetic trials by randomly sampling temporal segments across subjects to promote frequency-level invariance), channel chunk shuffling (simulating varied electrode placements by swapping channel chunks), random temporal cropping (addressing inter-subject latency variability), and channel decorrelation via covariance-based whitening to reduce subject-level variability. Training follows a four-stage procedure with staged transfer learning: ViT generative pretraining on all subjects, ViT subject-specific fine-tuning, MoE decoder pretraining using pooled data across subjects with augmentation, and MoE subject-specific fine-tuning for individual adaptation.
The experimental evaluation demonstrates substantial performance improvements over both traditional methods (CCA, TRCA, TDCA, eTRCA, eCCA, msTRCA) and deep learning baselines (DNN, SSVEPformer, TRCANet) on two large benchmark datasets spanning 105 subjects: the Benchmark dataset (35 subjects, controlled laboratory environment) and BETA dataset (70 subjects, naturalistic settings). At the shortest data length of 0.2 seconds, VIBE achieves the largest accuracy advantage (Benchmark: 65.5% vs. 58.8% for next-best; BETA: 53.7% vs. 46.2%), with maximum information transfer rates of 263.8±11.7 bpm at 0.4s for Benchmark and 202.7±8.9 bpm for BETA, exceeding DNN baseline by approximately 15 bpm and 12 bpm respectively. Statistical analysis via two-way repeated-measures ANOVA reveals significant interaction effects between method and data length (p<0.001 for both datasets), with paired t-tests confirming VIBE's superiority over DNN and TDCA across all evaluated time windows (all p<0.05), particularly pronounced at short durations (0.2s: p=1.0×10^-12 vs. DNN, p=2.2×10^-14 vs. TDCA for Benchmark). Comprehensive ablation studies quantify individual component contributions: removing ViT regeneration causes the largest performance drop to 61.8%, removing MoE reduces accuracy to 64.2%, and eliminating all data augmentation decreases performance to 62.1% from the full model's 65.5% on Benchmark at 0.2s. Further analysis shows that MoE placement on the second temporal layer is most effective, decorrelation has the strongest impact among augmentation strategies (63.3% without it), and optimal ViT generation time length is data-length dependent (longer generation benefits short trials, shorter generation for longer trials). t-SNE visualization reveals that the full model produces more compact and discriminative feature clusters compared to ablated versions, while subject-wise ITR radar plots demonstrate VIBE's consistent superiority across individual subjects, establishing its effectiveness as a high-performance decoding strategy for SSVEP-BCIs.

**Strengths:**

This paper presents a well-motivated approach to SSVEP-BCI decoding with several notable strengths. The proposed VIBE framework demonstrates clear architectural innovation by combining ViT-based temporal generation with MoE-based decoding in a manner specifically tailored to EEG signal characteristics, moving beyond generic application of existing architectures. The experimental validation is comprehensive and rigorous, employing proper cross-validation protocols (k=6 for Benchmark, k=4 for BETA) across two large-scale datasets spanning 105 subjects in both controlled laboratory and naturalistic settings, with consistent performance gains across all evaluated conditions. The results are particularly compelling at short time windows (0.2s), where VIBE achieves substantial accuracy improvements (6.7% on Benchmark, 7.5% on BETA) that directly address the stated challenge of rapid decoding. The ablation studies are thorough and well-designed, systematically isolating the contribution of each component (ViT, MoE, augmentation strategies) and providing additional analyses on architectural choices (MoE layer placement, generation time length effects, individual augmentation impacts), which collectively validate the design decisions and demonstrate that improvements stem from the synergistic combination of components rather than any single element. The paper also provides thoughtful discussion of neural underpinnings for each proposed module, connecting design choices to physiological properties of SSVEP signals and EEG characteristics, while appropriately acknowledging limitations regarding inter-subject variability and the need for online validation.

**Weaknesses:**

The paper's experimental design and evaluation methodology raise several concerns that limit the generalizability and interpretability of the findings. First, while the authors claim superior performance, the evaluation protocol employs trial-based cross-validation rather than subject-based cross-validation, which is problematic given that the model undergoes subject-specific fine-tuning in its fourth training stage. This means the model has already seen and adapted to data from the test subject during fine-tuning, potentially leading to optimistic performance estimates that may not reflect true generalization to completely unseen users. The claim of "subject-independent evaluation protocol" is therefore misleading, as genuine subject-independence would require testing on subjects whose data was never used during any training stage, including fine-tuning. Second, the comparison with baseline methods appears unbalanced: traditional methods like TDCA and TRCA receive no equivalent multi-stage training or subject-specific adaptation, while deep learning baselines like DNN are not provided with the same data augmentation strategies or temporal generation mechanisms that VIBE employs. A fairer comparison would either provide all baselines with similar data augmentation and multi-stage training opportunities, or evaluate VIBE without these advantages to isolate the true architectural contributions. Third, the paper lacks critical analyses for practical deployment, including computational cost comparisons during training and inference, memory requirements for storing subject-specific models, and cross-dataset generalization experiments where models trained on one dataset are directly tested on another without any fine-tuning. The brief mention of training times in the appendix (e.g., 4180 seconds for decoder pretraining on Benchmark) suggests substantial computational overhead, but no comparison with baseline methods is provided to contextualize whether this cost is justified.

Beyond experimental concerns, the paper suffers from limited technical novelty and insufficient depth in several key aspects. The core components of VIBE are largely borrowed from existing architectures: ViT for temporal generation is a straightforward application of the standard Vision Transformer to EEG data with minimal adaptation beyond treating multi-band signals as image tensors, while MoE has been previously applied to EEG decoding tasks as acknowledged by the authors. The primary contribution appears to be the specific combination and staged training of these existing modules rather than fundamental architectural innovation, yet the paper does not adequately justify why this particular combination is theoretically well-suited for SSVEP decoding compared to alternative fusion strategies. The ablation study only compares against removing components entirely or using simple co-attention, without exploring other sophisticated fusion mechanisms from the multimodal learning literature. Furthermore, the data augmentation strategies, while diverse, are presented without sufficient justification: cross-subject temporal stitching may actually harm performance by mixing neural responses from different individuals with distinct physiological characteristics, yet no analysis is provided to verify whether this augmentation actually improves generalization or simply acts as noise injection. The physiological interpretations offered in the discussion are speculative and lack empirical validation through techniques such as attention weight visualization, frequency spectrum analysis of learned representations, or correlation with known neurophysiological markers of SSVEP responses. The paper also lacks insight into what patterns the MoE experts actually specialize in learning, which temporal or spectral features the ViT generation emphasizes, or whether the learned representations align with established neuroscience understanding of SSVEP generation mechanisms, making it difficult to assess whether the model's success stems from capturing meaningful neural dynamics or exploiting dataset-specific statistical regularities.

**Questions:**

Please see your weaknesses and I will adjust the final score based on your answers.

---

> ### Author Response · Authors · 2025-11-24
> **Response to Reviewer 3S4g [1/n]**
>
> **The paper's experimental design and evaluation methodology raise several concerns that limit the generalizability and interpretability of the findings.**
>
> Thank you for your valuable comments on the manuscript! In the following, we clarify the generalizability of the proposed approach and conduct additional experiments to address interpretability concerns. We have revised the manuscript point by point in line with your suggestions. We would be happy to address any further questions or comments you may have.
>
> Best regards,
>
> The Authors

---

> ### Author Response · Authors · 2025-11-24
> **Response to Reviewer 3S4g [2/n]**
>
> **First, while the authors claim superior performance, the evaluation protocol employs trial-based cross-validation rather than subject-based cross-validation, which is problematic given that the model undergoes subject-specific fine-tuning in its fourth training stage. This means the model has already seen and adapted to data from the test subject during fine-tuning, potentially leading to optimistic performance estimates that may not reflect true generalization to completely unseen users. The claim of "subject-independent evaluation protocol" is therefore misleading, as genuine subject-independence would require testing on subjects whose data was never used during any training stage, including fine-tuning.**
>
> We thank the reviewer for the question and apologize for the confusion. Our evaluation protocol is not **trial-wise cross-validation**. Instead, we follow the standard **individually calibrated setting**, which we clarify here and have now made explicit in the revised manuscript.
>
> In SSVEP-BCI, there are broadly two evaluation scenarios: (1) individually calibrated decoding, where subject-specific data are used for training, and (2) calibration-free decoding, where models aim to generalize across subjects without individual calibration. Because SSVEP signals exhibit substantial inter-subject variability, individually calibrated protocols are the de facto choice when the goal is to maximize high-speed BCI performance. For this reason, VIBE is explicitly designed for the individually calibrated setting.
>
> Concretely, our protocol follows the same training strategy as prior works: cross-subject pretraining followed by subject-specific finetuning, with cross-validation ensuring that neither training nor finetuning ever sees the test trials. This individually calibrated evaluation protocol is well-established in the SSVEP-BCI literature and does not conflate training and test data or artificially inflate performance. By contrast, calibration-free/subject-independent scenarios (e.g., sinusoidal-reference methods such as CCA/FBCCA/MEC/MSI, or transfer-learning approaches with cross-subject validation) constitute a **separate line of work** and are outside the scope of the present study. Therefore, we did not claim to use a subject-independent evaluation protocol.
>
> We believe the misunderstanding may stem from some ambiguous phrasing in the original manuscript. We have now revised and highlighted the relevant passages mainly in the Conclusion and Discussion (marked in red in the rebuttal version in Section 7 and 6.1) to explicitly state that our experiments are conducted under an individually calibrated protocol.
>
> The following revision has been made in Section 7 of the paper.
> “In this study, we proposed the Vision Transformer Based Expert network (VIBE), a multistage deep learning framework that integrates a ViT-MoE architecture and novel data enhancement approaches tailored for SSVEP data. By leveraging information from short-duration EEG recordings, VIBE learns effective and discriminative neurophysiological representations for individually calibrated SSVEP decoding.”
>
> The ambiguous statement in Section 6.1 has been phrased as follows.
> “This increased diversity of neural signals mitigates overfitting to individual trials and promotes more robust class-specific representations, ultimately improving the model’s generalization performance.”
>
> Best regards,
>
> The Authors

---

> ### Author Response · Authors · 2025-11-24
> **Response to Reviewer 3S4g [3/n]**
>
> **Second, the comparison with baseline methods appears unbalanced: traditional methods like TDCA and TRCA receive no equivalent multi-stage training or subject-specific adaptation, while deep learning baselines like DNN are not provided with the same data augmentation strategies or temporal generation mechanisms that VIBE employs. A fairer comparison would either provide all baselines with similar data augmentation and multi-stage training opportunities, or evaluate VIBE without these advantages to isolate the true architectural contributions.**
>
> We thank the reviewer for this thoughtful comment about the fairness of the comparisons.
>
> For traditional methods such as TDCA and TRCA, these approaches are formulated as matched-filter style statistical models with closed-form solutions. Their parameters are estimated directly from the available data and are not optimized via gradient-based training. As a result, there is no natural analogue of multi-stage training or cross-subject pretraining for these models: once the spatial filters are recomputed on subject-specific data, any information from previous “stages” is effectively overwritten. In practice, TDCA/TRCA are already subject-specific in the sense that their filters are estimated separately for each subject, and extending them with additional ad-hoc training stages would move them away from their standard, well-established formulations.
>
> For deep learning–based baselines, our goal in this work is not to propose a generic data augmentation or multi-stage training framework, but to develop a rapid, high-performance SSVEP-BCI decoding system that can achieve high information transfer rates (ITR) with short data lengths. To ensure a fair and interpretable comparison, we therefore follow the original training protocols recommended by the authors of the baseline (including DNN), without retrofitting them with our temporal generation modules or additional augmentation strategies. Modifying each baseline with our proposed components would effectively create new variants of those methods and blur the attribution of performance gains. Instead, we propose VIBE as a complete system and compare it against existing methods under their respective best-practice settings on the benchmark datasets, thereby demonstrating that VIBE significantly pushes the performance and ITR frontier in the standard SSVEP-BCI evaluation setup.
>
>
> Best regards,
>
> The Authors

---

> ### Author Response · Authors · 2025-11-24
> **Response to Reviewer 3S4g [4/n]**
>
> **Third, the paper lacks critical analyses for practical deployment, including computational cost comparisons during training and inference, memory requirements for storing subject-specific models, and cross-dataset generalization experiments where models trained on one dataset are directly tested on another without any fine-tuning. The brief mention of training times in the appendix (e.g., 4180 seconds for decoder pretraining on Benchmark) suggests substantial computational overhead, but no comparison with baseline methods is provided to contextualize whether this cost is justified.**
>
> We appreciate the reviewer’s concern regarding computational practicality. In the revised manuscript, we now include a dedicated **Model Size Comparison** section with Table (Table 9 in the manuscript)  to contextualize the computational footprint of VIBE relative to standard baselines.
>
> **Model size comparison: parameters and approximate memory (FP32).**
>
> | Method       | Parameters | Approx. Size (MB, FP32) |
> |--------------|-----------|------------------------|
> | DNN          | 0.18M     | 0.7                    |
> | TRCANet      | 0.18M     | 0.7                    |
> | SSVEPformer  | 9.26M     | 37.1                   |
> | Ours         | 1.18M     | 4.7                    |
>
> Although our framework incorporates both a ViT encoder and an MoE decoder, the resulting model remains lightweight: 1.17M parameters (~4.7 MB in FP32). In the field of computer vision, where both ViT and Convnet were invented,  it is standard for models to possess on the order of 10–100 million parameters. This is sufficiently small for low-resource deployment, and thus the memory overhead is negligible in practice.
>
> We made the following revision in Section A.6 of the paper. ”We report both the total number of trainable parameters and the corresponding memory usage in FP32 precision. As summarized in Table, all models remain lightweight, with memory requirements well within the range suitable for real time or embedded deployment. For this analysis, we only consider the deep learning–based models. Traditional methods are excluded because they do not maintain persistent trainable parameters and their memory footprint is dominated by temporary data buffers rather than model weights, making a direct comparison of “model size” with neural architectures not meaningful. Note that TRCANet and the DNN share the same decoder architecture, resulting in exactly the same number of parameters. This comparison highlights the balance between model capacity and efficiency across the evaluated architectures. "
>
>
> As reported in the following Table (Table 7 in the manuscript ), while training the ViT and Decoder may seem computationally intensive—taking ~1.2 hours for the Benchmark dataset—the required time for the Beta dataset is only about one-third of that. Importantly, at test time, inference remains highly efficient, requiring less than 0.8 ms per sample, which is negligible compared to the actual data durations (0.2 s, 0.4 s, etc.).
>
> **Table. 7** Training and testing times for VIBE on two datasets for time data length 0.4 s. Times are in seconds, except for the test stages, which are in milliseconds.
>
> | Dataset   | ViT Train (s) | ViT Finetune (s) | ViT Test (ms) | MoE Train (s) | MoE Finetune (s) | MoE Test (ms) |
> |-----------|---------------|------------------|---------------|---------------|------------------|---------------|
> | Benchmark | 270.8         | 19.4             | 0.7           | 4180.3        | 43.2             | 0.09          |
> | BETA      | 328.9         | 5.7              | 0.7           | 1681.1        | 11.8             | 0.09          |
>
>
>
> Regarding dataset generalization, our study is designed around subject-specific decoding within a single dataset, which is the **dominant evaluation setting** in SSVEP-BCI research. Cross-dataset transfer is indeed valuable but introduces issues such as frequency mismatches, hardware variability, and different experimental paradigms, which is not the standard evaluation setting in the SSVEP-BCI research; thoroughly addressing these is beyond the scope of the current work. We plan to explore cross-dataset generalization in future extensions.
>
> Best regards,
>
> The Authors

---

> > ### Author Response · Authors · 2025-11-24
> > **Response to Reviewer 3S4g [9/n]**
> >
> > **Furthermore, the data augmentation strategies, while diverse, are presented without sufficient justification: cross-subject temporal stitching may actually harm performance by mixing neural responses from different individuals with distinct physiological characteristics, yet no analysis is provided to verify whether this augmentation actually improves generalization or simply acts as noise injection. The physiological interpretations offered in the discussion are speculative and lack empirical validation through techniques such as attention weight visualization, frequency spectrum analysis of learned representations, or correlation with known neurophysiological markers of SSVEP responses.**
> >
> > To address this concern, we have added a new subsection titled “Spectral Validation of Stitch-Augmented Data” (Sec. A.9 in the revision). In this analysis, we explicitly evaluate whether the Cross-Subject Temporal Stitching procedure preserves the characteristic frequency structure of SSVEP signals. After Stage-2 ViT sample generation and Stitch recomposition, we extract a single subband, average across channels and trials, zero-pad the resulting sequence to 5 seconds, and apply an FFT to obtain interpretable spectral profiles. As shown in Fig. 9, the stitched samples consistently exhibit clear spectral peaks at the target frequency and its harmonics, matching the behavior of authentic SSVEP signals. The corresponding temporal traces likewise show stable periodic structure aligned with the stimulus frequency. Although Stitch mixes temporal chunks across subjects, this controlled variability increases the effective training diversity for the decoder without **disrupting the frequency-specific signatures required for correct classification**. These results demonstrate that the Stitch-augmented instances retain the core SSVEP structure and therefore provide valid, neurophysiologically meaningful training examples.
> >
> > A section A.9 has been updated in the paper.
> > ‘To assess whether the Cross-Subject Temporal Stitching augmentation preserves useful SSVEP structure, we visualize the temporal and spectral profiles of the augmented samples. After Stage 2 ViT generation and Stitch recomposition, we select a single subband, average across channels and trials dimensions, zero-pad the resulting sequence to 5s, and apply an FFT. The corresponding temporal traces and spectra consistently exhibit clear peaks at the target frequency and its harmonic, indicating that the stitched signals retain the essential SSVEP signatures. Although Stitch mixes temporal chunks across channels and subjects, this variability diversifies the training distribution for the decoder while maintaining the frequency-specific structure required for accurate decoding. The accompanying plot (Fig. 9) contains two subpanels: the left panel shows the stitched time-series signal, and the right panel shows the FFT magnitude spectrum. In the spectral panel, clear peaks appear at the target frequency and its harmonic (2$\times$freq), confirming that the stitched samples preserve the characteristic SSVEP structure.’
> >
> >
> > Best regards,
> >
> > The Authors

---

> ### Author Response · Authors · 2025-11-24
> **Response to Reviewer 3S4g [5/n]**
>
> **Beyond experimental concerns, the paper suffers from limited technical novelty and insufficient depth in several key aspects. The core components of VIBE are largely borrowed from existing architectures: ViT for temporal generation is a straightforward application of the standard Vision Transformer to EEG data with minimal adaptation beyond treating multi-band signals as image tensors, while MoE has been previously applied to EEG decoding tasks as acknowledged by the authors.**
>
> We acknowledge that transformer structures have been applied to EEG in prior work, but these models almost universally operate as **feature extractors**, not generators. Existing ViT-based EEG architectures treat the transformer as an decoder that aggregates spatial or spatiotemporal information into a latent representation and to do classification . Even in recent EEG foundation models, transformers play the role of **token encoders**, **analogous to BERT tokenizers,where the goal is to compress signals into embedding spaces**, not to expand or synthesize representations. In contrast, our work is, to our knowledge, the first to employ a ViT **as a temporal generator** that extends short time-length EEG signals into longer effective representations. This design is not a trivial use of ViT: It directly tackles a fundamental challenge in SSVEP decoding by enhancing performance with longer temporal windows. However, in real-time scenarios, acquiring longer segments inevitably slows down decoding, as longer time series require more time to collect.. By generating extended temporal structure from shorter inputs, our ViT module introduces a functionality (temporal-length expansion) that no previous EEG transformer model supports. This constitutes a meaningful technical novelty beyond existing transformer-EEG applications.
>
> **Our MoE usage is functionally distinct from previous MoE-EEG work**. While EvoMoE introduced MoE for EEG, their objective is fundamentally different: they use large MoE blocks at the entire decoder level to discover new biological data distribution patterns. In contrast, we introduce a localized MoE inserted at lightweight Conv2d layers, whose purpose is **not biological pattern discovery**, but **frequency-specialized expert routing** tuned to SSVEP harmonic structure. This is a new operational role for MoE that has not appeared in prior EEG literature. Additionally, by limiting MoE to small layers, our design maintains computational efficiency while enabling fine-grained specialization, a different architectural philosophy from EvoMoE’s full-decoder expert banks.
>
> Together, the ViT-based temporal generator and the frequency-specialized, lightweight MoE form an architectural contribution that is not a trivial reuse of existing components but a novel adaptation specifically motivated by the constraints and physiology of SSVEP decoding.
>
> Best regards,
>
> The Authors

---

> ### Author Response · Authors · 2025-11-24
> **Response to Reviewer 3S4g [6/n]**
>
> **The primary contribution appears to be the specific combination and staged training of these existing modules rather than fundamental architectural innovation, yet the paper does not adequately justify why this particular combination is theoretically well-suited for SSVEP decoding compared to alternative fusion strategies.**
>
> We thank the reviewer for raising concerns regarding the justification of our staged design. In the revised manuscript, we clarify both the theoretical grounding and the empirical necessity of each stage. While the reviewer suggests that our approach merely combines existing components, our design is in fact tailored to the frequency-locked property of SSVEP signals, where preserving and enhancing harmonics is essential for discrimination. The four stages of VIBE are purposefully constructed to (1) learn subject-agnostic temporal regularities (Stage 1), (2) adapt these representations to subject-specific spectral patterns (Stage 2), (3) decode these signals (Stage 3), and (4) further refine decoding with subject-level fine-tuning (Stage 4).
>
> To address the reviewer’s request for deeper justification, we have added a new subsection in the revision: “**Evaluating the Contribution of All Stages via MoE Temporal Layer Ablation**.” This analysis examines how different stages influence the learned temporal–spectral structure using FFT-based SNR measured directly on the MoE temporal layer activations. The controlled comparison tests four conditions, Stage 3 vs. Stage 4 decoders and ViT-augmented vs. original-only input, while holding data and architecture constant.
>
> The updated results (Fig. 6 and Table 10 in the revision) show that both components criticized as insufficiently motivated: the ViT generator (Stages 1–2) and the final fine-tuning stage (Stage 4) produce **statistically significant improvements** in frequency-domain structure. Specifically, target-frequency SNR increases consistently from BASE-original → BASE-full → FT-original → FT-full, with all paired comparisons reaching statistical significance. These results demonstrate that each stage makes a measurable, nonredundant contribution aligned with the physiological structure of SSVEPs.
>
> The paper has been updated as follows in Section A.7.
>
> ”In this analysis, we investigate how different stages of VIBE influence the model’s frequency domain behavior under a controlled experimental setting. We use slightly modified parameters relative to the main experiments: each input consists of a 1s EEG segment, and the Stage 1 ViT generator produces an additional 0.2s extension. The decoder is implemented with eight temporal experts in the MoE layer. Both the Stage 3 (non fine tuned) and Stage 4 (finetuned) decoders are trained using the ViT augmented data, and all visualizations are performed using the same training set to remove cross-split variability. For each subject and trial, we extract the activation from the final temporal Conv2D layer and average across channels as well as subjects to obtain a single temporal sequence per target. This sequence is zero-padded to 5s to achieve sufficient spectral resolution for targets ranging from 8 Hz to 15.8 Hz, after which we apply an FFT. We compute the signal-to-noise ratio (SNR) by taking the peak amplitude at the target frequency and its second harmonic, normalizing each by the mean amplitude of neighboring frequency bins, and averaging the two values. We then perform paired t-tests to assess statistical significance. The analysis compares four conditions: (1) Stage 3 versus Stage 4 decoders, and (2) ViT augmented input (original+generated) versus original only input with zero-padding. The corresponding FFT visualizations and t-test statistics are summarized in the accompanying Figure 6  and Table 10.
>
> The FFT visualizations and t-test results consistently show that both the inclusion of ViT-generated data and decoder fine-tuning improve spectral responses. Specifically, the amplitude at the target frequency and its second harmonic is highest for FT with full input, followed by FT with original input, then BASE full, and finally BASE original. All comparisons are statistically significant, highlighting that ViT augmentation and Stage 4 fine-tuning are important for enhancing frequency-specific features in the model."
>
> Together, this new analysis provides a direct, quantitative justification of our staged architecture and shows that the design is not an arbitrary combination but a sequence of transformations that progressively enhances harmonic fidelity, whereas something alternative fusion strategies do not guarantee. Accordingly, we believe the revised manuscript now clearly motivates why VIBE’s staged composition is theoretically appropriate and empirically necessary for high-accuracy SSVEP decoding.
>
>
> Best regards,
>
> The Authors

---

> ### Author Response · Authors · 2025-11-24
> **Response to Reviewer 3S4g [7/n]**
>
> **The primary contribution appears to be the specific combination and staged training of these existing modules rather than fundamental architectural innovation, yet the paper does not adequately justify why this particular combination is theoretically well-suited for SSVEP decoding compared to alternative fusion strategies.**
>
> **Table 10**. Paired t-test results comparing SNR differences from the final temporal layer activations.
> FT refers to Stage 4 and BASE refers to Stage 3. *Full* uses the complete input sequence (original data + Stage 2 ViT-generated data), while *Original* keeps only the original data and zero-pads the rest.
>
> | Comparison                         | t-stat  | p-value               | n  |
> |------------------------------------|--------:|-----------------------|:--:|
> | FT (full $-$ original)             | 3.3941  | $1.59\times10^{-3}$   | 40 |
> | BASE (full $-$ original)           | 3.0741  | $3.84\times10^{-3}$   | 40 |
> | FT full $-$ BASE full              | 9.2720  | $2.08\times10^{-11}$  | 40 |
> | FT original $-$ BASE original      | 7.6147  | $3.12\times10^{-9}$   | 40 |
>
> Best regards,
>
> The Authors

---

> ### Author Response · Authors · 2025-11-24
> **Response to Reviewer 3S4g [8/n]**
>
> **The ablation study only compares against removing components entirely or using simple co-attention, without exploring other sophisticated fusion mechanisms from the multimodal learning literature.**
>
> We sincerely appreciate the reviewer’s suggestion regarding the exploration of sophisticated multimodal fusion mechanisms. However, we would like to clarify that our study exclusively uses a single data modality, EEG signals, without incorporating any additional sources of information. Consequently, multimodal fusion techniques are not applicable in this context. The ablation study is therefore designed to evaluate the contributions of components within the single-modality framework, which is the most appropriate approach for our work.
>
>
> Best regards,
>
> The Authors

---

> ### Author Response · Authors · 2025-11-24
> **Response to Reviewer 3S4g [10/n]**
>
> **The paper also lacks insight into what patterns the MoE experts actually specialize in learning, which temporal or spectral features the ViT generation emphasizes, or whether the learned representations align with established neuroscience understanding of SSVEP generation mechanisms, making it difficult to assess whether the model's success stems from capturing meaningful neural dynamics or exploiting dataset-specific statistical regularities.**
>
> To address this concern, we added a qualitative interpretability analysis focusing on the MoE temporal layer in the fine-tuned decoder. Specifically, we apply Grad-CAM [1] to the final MoE temporal convolution layer to obtain expert-relevant temporal saliency. The temporal map is averaged across channels, zero-padded to 5 seconds for consistent spectral resolution, and analyzed in both the time and frequency domains via FFT. In parallel, we record expert-selection counts for each target frequency to measure how the MoE distributes responsibility across experts.
>
> The results show that the MoE does **not** collapse to a single expert; instead, different target frequencies consistently elicit **distinct expert-activation profiles**, indicating that the experts specialize in different temporal–spectral structures rather than memorizing redundant patterns. The Grad-CAM spectral analysis further reveals **clear peaks at the stimulus frequency and its harmonic**, demonstrating that the learned representations indeed track canonical SSVEP signatures rather than dataset artifacts. Together, these analyses provide direct evidence that the MoE architecture decomposes the temporal dynamics into meaningful components and that the model’s success stems from capturing **neurophysiologically grounded SSVEP structure**, not from exploiting spurious statistical regularities.
>
> The following revision has been made in the Section A.8 in the paper.
>
> “We perform a qualitative analysis of the MoE temporal layer using the fine-tuned decoder for a randomly selected subject. Grad-CAM [1] is applied to the final MoE temporal convolution layer to obtain activation importance over time. The resulting temporal map is averaged across channels to produce a single 1D sequence. As in the previous analyses, this sequence is zero-padded to 5s to ensure sufficient spectral resolution before applying the FFT. In addition to the frequency-domain visualization, we record the MoE expert selection count for each target frequency to examine how the mixture-of-experts distributes attention across different spectral components.
> The MoE expert selection patterns reveal that the experts do not collapse into a single dominant expert; instead, different target frequencies elicit distinct expert activation distributions. This diversity indicates that individual experts specialize in different temporal–spectral patterns rather than redundantly modeling the same structure. Complementing this, the Grad-CAM analysis shows clear spectral peaks at the target frequency and its harmonic after FFT, demonstrating that the MoE temporal layer effectively pools and amplifies frequency-specific structure. Together, these results confirm that the MoE architecture meaningfully decomposes the temporal dynamics and contributes specialized processing across targets.”
>
> [1] Selvaraju et al. Grad-cam: Visual explanations from deep networks via gradient-based localization. ICCV, 2017.
>
>
> Best regards,
>
> The Authors

---

> ### Comment · Reviewer_3S4g · 2025-11-25
> **Thank you for your reply. I will significantly improve my score.**
>
> thanks again

---

> > ### Author Response · Authors · 2025-11-28
> > **Thank you for your reply**
> >
> > Dear Reviewer,
> >
> > Thank you for the update. I appreciate your time and constructive feedback. Please let me know if there is anything else I can clarify. We have carefully addressed all points raised, and the revised submission now reflects all corresponding updates and clarifications. Please feel free to provide any additional feedback or suggestions. We appreciate your time and effort in helping us improve our work.
> >
> > Best regards,
> >
> > The Authors

---

### Official Review · Reviewer_bXb6 · 2025-11-01

**Soundness:** 3
**Presentation:** 3
**Contribution:** 3
**Rating:** 6
**Confidence:** 2

**Summary:**

The paper introduces VIBE (Vision Transformer-Based Experts Network), a deep learning framework designed for SSVEP-based BCI decoding. VIBE employs a Vision Transformer (ViT) for temporal signal extension, complemented by a Mixture-of-Experts (MoE) decoder to exploit subband, channel, and temporal dependencies. Novel EEG-specific augmentations (temporal stitching, channel chunk shuffle, decorrelation) enhance model generalization. Evaluations on Benchmark and BETA datasets (105 subjects total) demonstrate superior performance, consistently surpassing traditional (e.g., eCCA, TRCA, TDCA) and deep (DNN, SSVEPformer) baselines.

**Strengths:**

- The combination of ViT temporal generation and specialized MoE decoder is novel and thoughtfully justified. EEG-specific augmentations are creative and practically useful.
- Comprehensive experimentation on large datasets and strong baselines clearly demonstrate performance improvements, supported by robust ablation analyses.
- Well-described methodology, clear visualization (architecture, accuracy/ITR plots, t-SNE feature visualization).
- Substantial accuracy and ITR gains for short observation periods demonstrate clear potential for practical high-speed BCIs.

**Weaknesses:**

- Lack of cross-subject validation limits claims about broader applicability. Evaluations are confined to within-subject validation, limiting generalization conclusions.
- The realism of channel chunk shuffle is not sufficiently evaluated. Alternative realistic augmentation methods are not explored.
- Assumed gaze-shift times might differ in practical scenarios; online evaluation would validate practical effectiveness.

**Questions:**

- Could you provide leave-one-subject-out (LOSO) evaluations to assess generalization robustness clearly?
- Statistical Clarification: Were multiplicity corrections (Holm/FDR) applied to paired t-tests? Please clarify.
- Electrode Sensitivity: How sensitive is performance to variations in electrode number or placement?

---

> ### Author Response · Authors · 2025-11-26
> **Response to Reviewer bXb6 [1/n]**
>
> **Weakness 1. Question 1. Lack of cross-subject validation limits claims about broader applicability. Evaluations are confined to within-subject validation, limiting generalization conclusions. Could you provide leave-one-subject-out (LOSO) evaluations to assess generalization robustness clearly?**
>
> We thank the reviewer for the suggestion.However, in SSVEP-BCI research, it is standard to evaluate either in a cross-subject (LOSO) or an individually calibrated (LOBO) setting, **where most works choose one, not both**. Our goal is to achieve maximal accuracy and ITR, which is only attainable in an individually calibrated setting. This is why the field’s high-speed SSVEP-BCI literature overwhelmingly adopts LOBO  evaluation rather than LOSO when reporting state-of-the-art performance.
> Both datasets used in our work include a total of 105 subjects, already providing strong coverage of inter-subject variability. Following the standard LOBO protocol, which is established in prior works on the Benchmark and BETA datasets, ensures direct comparability and reflects the practical deployment scenario where per-subject calibration is required to maximize BCI speed and accuracy.
> While LOSO is relevant for universal decoders, it serves a different goal. Our method is explicitly designed for the individually calibrated, high-performance regime, where LOBO is the appropriate and standard evaluation strategy.
>
> **Weakness 2. The realism of channel chunk shuffle is not sufficiently evaluated. Alternative realistic augmentation methods are not explored.**
>
> We thank the reviewer for the comment. EEG channels exhibit distinct spectral characteristics but do not follow a strict sequential order like RGB channels in images, although some channels are more dominant. To enhance model generalization, we apply shuffling within defined temporal-spatial chunks, exposing the network to different channel arrangements during training. This augmentation preserves the underlying physiological information, as the network learns invariant spectral patterns rather than relying on fixed channel positions. To further address this concern, we added Section A.10, “Spectral Validation of Channel Chunk Shuffle Augmented Data,” which demonstrates that the shuffled samples retain their characteristic frequency structures.
>
> "To verify that the Channel Chunk Shuffle augmentation preserves essential SSVEP structure, we visualized the spectral profiles of augmented samples. We use 1 s temporal length original data as input from Benchmark. Following Stage 2 ViT generation (add 0.5s data) and shuffle recomposition, we select a single subband and channel, average across trials, zero-pad the resulting sequence to 5 s, and compute the FFT.
> By shuffling within defined temporal chunks, we expose the network to different channel arrangements, enhancing generalization while preserving physiologically meaningful spectral features. The accompanying plot (Figure) shows that prominent peaks at the target frequency and its harmonic, indicating that the shuffled signals maintain the frequency-specific structure required for accurate decoding."
>
> **Weakness 3. Assumed gaze-shift times might differ in practical scenarios; online evaluation would validate practical effectiveness.**
>
> We agree that gaze-shift and stimulus-locking latencies may vary across individuals in practical systems. Such variability has been extensively documented in the SSVEP literature [1 - 7].
> Therefore, our proposed Random Temporal Crop augmentation directly addresses this issue by enhancing robustness to individual latency differences. For practical deployment, we plan to include an online evaluation in future work to validate the real-time effectiveness of the method under natural, variable gaze-shift conditions.
>
> [1]Liu, B. et al.  Improving the performance of individually calibrated SSVEP-BCI by task-discriminant component analysis. IEEE Transactions on Neural Systems and Rehabilitation Engineering 2021
>
> [2]Wang et al. A benchmark dataset for SSVEP-based brain–computer interfaces. IEEE Transactions on Neural Systems and Rehabilitation Engineering 2016
>
> [3]Pan et al. Enhancing the classification accuracy of steady-state visual evoked potential-based brain–computer interfaces using phase constrained canonical correlation analysis. Journal of neural engineering 2011
>
> [4]Lemm et al. Spatio-spectral filters for improving the classification of single trial EEG. IEEE transactions on biomedical engineering 2005
>
> [5]Dornhege et al.  Combined optimization of spatial and temporal filters for improving brain-computer interfacing. IEEE transactions on biomedical engineering 2006
>
> More citation continued next

---

> ### Author Response · Authors · 2025-11-26
> **Response to Reviewer bXb6 [2/n]**
>
> [6]Wu et al.  Classifying single-trial EEG during motor imagery by iterative spatio-spectral patterns learning (ISSPL). IEEE Transactions on Biomedical Engineering  2008
>
> [7]Qi et al. RSTFC: A novel algorithm for spatio-temporal filtering and classification of single-trial EEG. IEEE transactions on neural networks and learning systems  2015
>
> **Question 2. Statistical Clarification: Were multiplicity corrections (Holm/FDR) applied to paired t-tests? Please clarify.**
>
> We thank the reviewer for raising this point. We update this in the revision. “After applying the Holm-Bonferroni correction to control for multiple comparisons, all paired t-test results remained statistically significant (p < 0.05), indicating that VIBE consistently outperforms the compared methods across data lengths and datasets.”
>
> **Question 3. Electrode Sensitivity: How sensitive is performance to variations in electrode number or placement?**
>
> We thank the reviewer for raising this point. We systematically evaluated the impact of electrode number and placement by testing our model on five commonly used montages, ranging from 3 to 64 channels (Table). Our results show that classification accuracy generally improves as more channels are used, peaking at 70.2% with 30 channels. Using all 64 channels does not further increase accuracy and may reduce system practicality. Importantly, our model still achieves competitive performance with only 9 channels (classical occipital montage), demonstrating robustness to electrode reduction.
>
> We added a new section  A.13 in appendix named Electrode Configuration and Sensitivity.
>
> "The choice of EEG electrodes can significantly impact both the performance and practicality of BCI systems. Different electrodes capture overlapping but distinct spatial patterns of brain activity, and their number and placement can influence classification accuracy. To evaluate this, we tested our model using several commonly adopted electrode sets []:
> Central occipital montage (Nch = 3): Oz, O1, O2
> Classical occipital montage (Nch = 9):} Pz, POz, PO3/4, PO5/6, Oz, O1/2}
> Occipital montage (Nch = 21):} Pz, P1/2, P3/4, P5/6, P7/8, POz, PO3/4, PO5/6, PO7/8, Oz, O1/2, CB1/2}
> Parietal-occipital montage (Nch = 30): CPz, CP1/2, CP3/4, CP5/6, TP7/8, Pz, P1/2, P3/4, P5/6, P7/8, POz, PO3/4, PO5/6, PO7/8, Oz, O1/2, CB1/2
> Full montage (Nch = 64):} all channels
>
>
>
> Our results show that classification accuracy improves as the number of EEG channels increases from 3 to 30, reaching a peak of 70.2$\%$ with 30 channels. However, using all 64 channels does not further improve performance and may reduce system practicality. Considering the trade-off between accuracy and usability, we select a 9-channel configuration (classical occipital montage) for the experiments in this study.
> }
> Benchmark classification accuracy (\%) for different numbers of EEG channels.
>
> | Nch | Benchmark Acc (%) |
> |-----|-----------------|
> | 3   | 39.7            |
> | 9   | 65.5            |
> | 21  | 68.7            |
> | 30  | 70.2            |
> | 64  | 63.6            |
>
> [1]Liu, B. et al. Improving the performance of individually calibrated SSVEP-BCI by task-discriminant component analysis. IEEE Transactions on Neural Systems and Rehabilitation Engineering 2021
>
> Best regards,
>
> The Authors

---

> ### Author Response · Authors · 2025-11-28
> **Response to Reviewer bXb6 [3/n]**
>
> Dear Reviewer bXb6,
>
> Thank you for your thoughtful and constructive comments. We have carefully addressed all points raised, and the revised submission now reflects all corresponding updates and clarifications. Please feel free to provide any additional feedback or suggestions. We appreciate your time and effort in helping us improve our work.
>
> Best regards,
>
> The Authors

---

> > ### Comment · Reviewer_bXb6 · 2025-11-28
> >
> > Thank you for the response. The rebuttal has successfully resolved my concerns. I will maintain my positive score.

---

### Author Response · Authors · 2025-12-01
**Global Response and Revisions Summary**

Dear Reviewers, AC, SAC, and PC,

We sincerely thank all reviewers for their time, constructive feedback, and highly engaging discussion throughout the rebuttal period. We are grateful for the careful examination of our work and for the many positive assessments we received regarding its novelty (3S4g, ideL), clarity of the experimental setup paradigm (bXb6, 3S4g), comprehensive evaluation results (S2CM, ideL), additional ablations and visualizations (ideL, 3S4g, bXb6, S2CM), presentation quality (S2CM), and theoretical soundness (S2CM, 3S4g, bXb6). We have addressed all practical suggestions and provided clarifications for every comment.
- **We have added additional ablation studies:**
  - Following ideL, we conducted an ablation on the Comparison of Generators for Temporal Sequence Extension (Appendix K).
  - Following ideL and 3S4g, we analyzed the Multi-Stage Contributions with MoE (Appendix G).
  - Following bXb6, we performed more detailed ablations on Electrode Configuration and Sensitivity (Appendix L).

- **We have added additional visualizations and interpretability analyses:**
  - Following ideL and 3S4g, we added a Visualization Analysis for the MoE Temporal Layer (Appendix H).
  - Following S2CM, 3S4g, and bXb6, we conducted Spectral Validation of Data Augmentation (Appendix I.1 and I.2).
  - Following S2CM, we added another Data Augmentation ablation table (Appendix D.2), showing the effect of progressively adding augmentation components.

- **We have added methodological rationale and model details:**
  - Following bXb6, we clarified Section 3.3 on Random Temporal Crop, explaining gaze-shift considerations.
  - Following bXb6, we updated Section 5 to include Holm-Bonferroni correction for multiple comparison in t-tests.
  - Following 3S4g, we added a Model Size Comparison (Appendix B).
  - Following S2CM and ideL, we included Comprehensive Result Tables and Additional Baselines (Appendix J).

- **We have also improved the wording throughout the manuscript to explicitly clarify the experimental setup, removing any ambiguity about the leave-one-subject-out (LOSO) and individually calibrated evaluation paradigms.**

Overall, we believe these revisions have strengthened the clarity, rigor, and completeness of our work. We sincerely hope that the updated manuscript addresses all concerns and demonstrates the significance and robustness of our contributions.


Best regards,

The Authors

---

### Meta-Review · Area_Chair_avJ6 · 2026-01-07

**Summary:**

This paper proposes VIBE, a ViT- and MoE-based architecture for high-speed SSVEP decoding, coupled with EEG-specific augmentation and staged training. The reviewers raised concerns regarding evaluation protocol clarity, fairness of comparisons, augmentation validity, computational practicality, and architectural novelty. The rebuttal clarified the intent and correctness of the individually calibrated setting, added comprehensive numerical results and ablations, provided FFT-based validation of augmentation, characterized MoE specialization and model behavior, and supplied model size and runtime analysis. However, several concerns remain insufficiently resolved. In particular, there are still issues regarding the clarity and completeness of the evaluation protocol, the strength and statistical significance of the reported gains, and the motivation and practical benefit of the MoE-based architectural choices. Some ablation and rebuttal claims are not fully supported by the revised results, and presentation should be further improved. Thus, I recommend rejection. The authors could substantially improve this work and submit a much stronger version in the future.

**Reviewer Concerns:**

Reviewer bXb6 questioned the lack of cross-subject validation, the plausibility of channel shuffle augmentation, statistical testing, and electrode dependence. The rebuttal clarified that individually calibrated evaluation is standard, added spectral validation, provided corrected statistics and electrode analysis, and acknowledged online tests as future work. The reviewer confirmed all concerns were resolved.

Reviewer 3S4g raised concerns, including potential evaluation misinterpretation, unfair baseline comparison, limited novelty, unclear augmentation validity, and missing interpretability. The rebuttal clarified the protocol, justified baseline choices, added model-size and timing data, provided FFT and Grad-CAM analyses, and explained architectural roles. The reviewer significantly upgraded the score.

Reviewer S2CM requested complete numeric results, richer ablations, clear hyperparameter rationale, and justification for augmentation. The rebuttal supplied full tables, expanded ablations, clarified settings, and added spectral support. The response were satisfied.

For Reviewer ideL, the concerns include high-level motivation with limited context, an incomplete related-work section lacking recent EEG/BCI papers, and uncertainty about the novelty and justification of combining ViT-based temporal generation with an MoE layer. The rebuttal expands the literature review, adds more baselines and results tables, includes ablations on temporal generators and multi-stage MoE contributions, and adds visualizations to clarify design choices. These additions seem to address ideL’s concerns.

**Reviewer Scores:**

Reviewer bXb6 stated that the rebuttal resolved the concerns and that the reviewer would keep a positive evaluation.

Reviewer 3S4g confirmed that the authors’ responses significantly addressed the concerns and indicated the reviewer would substantially raise the score, moving from rejection into a clear acceptance position.

Reviewer S2CM did not explicitly report a new score.

The key issues in Reviewer ideL's reviews were addressed through expanded context, baselines, and justification.

---

### Decision · Program_Chairs · 2026-01-26

Reject